# Peer review analyze: A novel benchmark resource for computational analysis of peer reviews

**Tirthankar Ghosal**[1] *****, **Sandeep Kumar**[2], **Prabhat Kumar Bharti**[2], **Asif Ekbal**[2]

**1** Institute of Formal and Applied Linguistics, Faculty of Mathematics and Physics, Charles University, Prague, Czech Republic, **2** Indian Institute of Technology Patna, Bihta, Bihar, India

***** ghosal@ufal.mff.cuni.cz

**Data Availability Statement:** All the codes and data are available on GitHub (https://github.com/Tirthankar-Ghosal/Peer-Review-Analyze-1.0).

**Funding:** The first author thanks Visvesvaraya Ph. D. Fellowship Award (# VISPHD-MEITY-2518)

## Abstract

Peer Review is at the heart of scholarly communications and the cornerstone of scientific publishing. However, academia often criticizes the peer review system as non-transparent, biased, arbitrary, *a flawed process at the heart of science*, leading to researchers arguing with its reliability and quality. These problems could also be due to the lack of studies with the peer-review texts for various proprietary and confidentiality clauses. Peer review texts could serve as a rich source of Natural Language Processing (NLP) research on understanding the scholarly communication landscape, and thereby build systems towards mitigating those pertinent problems. In this work, we present a first of its kind multi-layered dataset of 1199 open peer review texts manually annotated at the sentence level ($\sim$ 17k sentences) across the four layers, *viz.* Paper Section Correspondence, Paper Aspect Category, Review Functionality, and Review Significance. Given a text written by the reviewer, we annotate: to which sections (e.g., Methodology, Experiments, etc.), what aspects (e.g., Originality/Novelty, Empirical/Theoretical Soundness, etc.) of the paper does the review text correspond to, what is the role played by the review text (e.g., appreciation, criticism, summary, etc.), and the importance of the review statement (major, minor, general) within the review. We also annotate the sentiment of the reviewer (positive, negative, neutral) for the first two layers to judge the reviewer's perspective on the different sections and aspects of the paper. We further introduce four novel tasks with this dataset, which could serve as an indicator of the *exhaustiveness* of a peer review and can be a step towards the automatic judgment of *review quality*. We also present baseline experiments and results for the different tasks for further investigations. We believe our dataset would provide a benchmark experimental testbed for automated systems to leverage on current NLP *state-of-the-art* techniques to address different issues with *peer review quality*, thereby ushering increased transparency and trust on the *holy grail of scientific research validation*. Our dataset and associated codes are available at https://www.iitp.ac.in/$\sim$ai-nlp-ml/resources.html#Peer-Review-Analyze.

from the Ministry of Electronics and Information Technology (MEITY), Government of India. The fourth author is also a recipient of the Visvesvaraya Young Faculty Award and acknowledges Digital India Corporation, MEITY for supporting this research."

**Competing interests:** The authors have declared that no competing interests exist.

# 1 Introduction

The peer-review process is the only widely accepted method of research validation. However, academia often criticizes the peer review system as non-transparent [1, 2], biased [3], arbitrary [4] and inconsistent [5, 6], *a flawed process at the heart of science* [7], leading to researchers arguing with its reliability [8] and quality [9, 10]. The *meta-research* and *science of science* [11] communities have long invested in studying annals of the peer-review process [12–17]. Alarmed by some of these glaring problems at the core of the review process [18], coupled with the exponential increase in research paper submissions, the larger research community (not only just *Meta Science*) felt the need to study the paper-vetting system and build proposals towards mitigating its problems [19]. There have been efforts in several venues [20–26] to guide reviewers to write good reviews, curating detailed guidelines in reviewer forms, all aimed towards the objective and fair evaluation of submitted manuscripts, thereby restoring the faith in the widely accepted method of scholarly communication. However, the majority of these efforts, which are targeted towards reforms on reviewers, might not be sufficient as there is an implicit human bias to invest less time in this voluntary yet critical job [27]. Also, the exploding nature of paper submissions leads to paper-vetting by less-experienced researchers [28]. Additionally, the bias of the reviewers [29], leading to inconsistencies between their review reports [5] further aggravates the problem. The consequences are often not welcome, sometimes leading to good research being ignored [30], and occasionally sub-par investigations finding a place in the community [31] (e.g., the recent retractions of peer-reviewed critical COVID-19 research [32]). Thus ensuring *Quality of Peer Reviews* is a time-critical problem.

However, estimating *Peer Review Quality* is not straightforward. We need to understand the underlying nuances of the peer review texts and the reviewer's intent manifested in those texts. Natural Language Processing has made a lot of exciting progress in recent years [33]. We believe that the peer-review reports can serve as a rich treatise of NLP research to gain more insights into the review process, to study the various facets of reviews quantitatively, to build automated tools for assessing bias and inconsistencies in peer-reviews, and finally, to understand the dynamics associated with the paper-vetting system and the scholarly communications landscape in-general. We are interested to know how we can leverage the *state-of-the-art* methods to study publicly available reviews and draw meaningful insights over the critical problem of *review quality*.

Peer reviews are meant to be constructive feedback to the authors. The reviewers are expected to make constructive criticisms over certain crucial aspects and sections of the paper in a detailed manner. Hence, identifying whether the reviewer was detailed or *exhaustive* in their review and has covered major sections, aspects of the paper are important to assess how much importance the editor/chair should pay to the particular review in making a decision (Accept/Reject/Revise). However, this assertion is domain-specific and can cease to hold in domains like humanities and social sciences where reviewers sometimes touch in depth on very few sections and aspects only. Going through the standard guidelines for peer-reviewing in major venues [20–26], we learn that there are certain aspects that the community wants to see in a good peer review [34]. We assert that: *a good peer review should comment on the important sections, address the critical aspects of the paper, perform some definitive roles, while clearly bringing out the reviewer's stand on the work.*

Let us take the example of peer reviews about a fictional paper #42 https://drive.google.com/drive/folders/1YySOUHo5Ae5Efi33SF0ffP5dsIp6NEM_ from the ACL 2020 Tutorial on Reviewing Natural Language Processing Research [35] to understand the variation in the quality of the review reports. We annotate the three reviews with our annotation schema (which

Box 1: R1: Weak Accept (Layer 1 and 2)

This paper presents a new covariance function for Gaussian processes (GPs) that is equivalent to a Bayesian deep neural network with a Gaussian prior on the weights and an infinite width.[[INT-NEU,MET-NEU], [NOV-NEU]] As a result, exact Bayesian inference with a deep neural network can be solved with the standard GP machinery. [[MET-NEU]] Pros: The result highlights an interesting relationship between deep nets and Gaussian processes.[[RES-POS], [EMP-POS]] (Although I am unsure about how much of the kernel design had already appeared outside of the GP literature.) [[EXP-NEU]] The paper is clear and very well written.[[OAL-POS], [CLA-POS]] The analysis of the phases in the hyperparameter space is interesting and insightful.[[ANA-POS], [EMP-POS]] On the other hand, one of the great assets of GPs is the powerful way to tune their hyperparameters via maximisation of the marginal likelihood but the authors have left this for future work![[FWK-NEU], [IMP-NEU]] Cons: Although the computational complexity of computing the covariance matrix is given, no actual computational times are reported in the article.[[EXP-NEG], [EMP-NEG]] I suggest using the same axis limits for all subplots in Fig 3.[[TNF-NEU]] https://openreview.net/forum?id=B1EA-M-0Z.

we discuss later; kindly follow Tables 2 and 3 for details). The three reviews (R1, R2, R3) clarify which review is more detailed and provides better feedback to the authors. Along with length, one striking characteristic of R3 (see next page) is that it is *exhaustive*. R3's comments on major sections (see Table 2) of the paper address several aspects (see Table 2) and clearly brings out the reviewer's perspective on the work. R1 seems to criticize the work heavily, still does a weak accept. R2 is optimistic about the work but provides little evidence and is poles apart from R1 and R3. R3 is comparatively the most *exhaustive* review and clearly indicates that the reviewer has spent time and effort to craft the review. We agree that the review's length is an important signal to measure *review exhaustiveness*, but only the *length* would be a trivial baseline to this seemingly complex task.

We do not claim that an exhaustive review involuntarily means a *good quality* review, but we deem that *exhaustiveness* is one dimension towards ascertaining *review quality*.

Also, identifying the role of peer review statements i.e., the texts that reviewers write to comment on the paper under scrutiny during the reviewing phase within a peer review seems an important direction to probe. We identify that reviewers write review statements with specific purposes and the review statements play specific roles within the review: present a concise summary of the work to indicate their understanding of the paper, provide suggestions to the authors to improve on their article, highlight deficit/shortcomings or missing components/aspects of the paper, appreciate the work, criticize it, demand clarifications from the authors (ask questions), present their knowledge and insights while discussing the research under review, bring out their overall recommendation on the article, etc. (See Table 3). All these roles can help us computationally analyze the reviewer's perspective on the work. Going further, these roles can also be signals towards the quality of the peer review (*if the reviewer has paid critical attention while reviewing the paper, comments were constructive or not, whether the review follows a general review structure? etc.*).

We also want to identify the comments that are central to the review and influence the overall decision. Not all the comments are crucial. Some are just minor, and some are general

> ## Box 2: R2: Strong Accept (Layer 1 and 2)
>
> The authors of this paper propose some extensions to the Dynamic Coattention Networks models presented last year at ICLR.[[INT-NEU]] First they modify the architecture of the answer selection model by adding an extra coattention layer to improve the capture of dependencies between question and answer descriptions.[[MET-NEU]] The other main modification is to train their DCN+ model using both cross entropy loss and F1 score (using RL supervision) in order to reward the system for making partial matching predictions.[[MET-NEU]] Empirical evaluations conducted on the SQuAD dataset indicates that this architecture achieves an improvement of at least 3%, both on F1 and exact match accuracy, over other comparable systems.[[EXP-POS,RES-POS], [CMP-POS,EMP-POS]] An ablation study clearly shows the contribution of the deep coattention mechanism and mixed objective training on the model performance. [[MET-NEU,ANA-NEU], [EMP-NEU]] The paper is well written, ideas are presented clearly and the experiments section provide interesting insights such as the impact of RL on system training or the capability of the model to handle long questions and/or answers.[[EXP-POS,OAL-POS], [CLA-POS,IMP-POS,PNF-POS]] It seems to me that this paper is a significant contribution to the field of question answering systems. [[OAL-POS], [REC-POS]] https://openreview.net/forum?id=H1meywxRW.

discussions (See Table 3). Strongly opinionated comments are the ones that the area chairs would pay more attention to, and the authors would also like to address them in their subsequent versions. Knowing which comments are significant would also provide signals to estimate the reviewer's confidence, knowledge, and strength of the review. Also, identifying crucial statements could help the area chairs to draft a proper meta-review for the paper.

With all these objectives, we went on to study the open access publicly available reviews of the International Conference on Learning Representations (ICLR) https://iclr.cc/ on the open-review platform https://openreview.net/. Specifically, we seek to investigate:

- *On which section of the paper is the review text talking about?*

- *Which aspect of the paper is addressed in a given review text?*

- *What is the role of a review statement in the peer review?*

- *Is the statement crucial to the review?*

Our research differs from the earlier works in the way we prepare the data for specific NLP tasks on peer review texts motivated towards developing computational models for peer review quality. Although our dataset consists of the reviews from a premier machine learning conference, we believe that our investigations would represent a general trivia of peer review in Science, Technology, Engineering, and Mathematics (STEM) disciplines.

To facilitate such studies, we propose a novel dataset of peer-review reports (1199 reviews) from the ICLR 2018 conference annotated mostly at the sentence level ($\sim$17k sentences) across four layers. The four layers are:

- **Layer 1: Review-Paper Section Correspondence**

- **Layer 2: Review-Paper Aspect**

## Box 3: R3: Strong Reject (Layer 1 and 2)

The below review addresses the first revision of the paper[[EXT-NEU]]. The revised version does address my concerns.[[OAL-POS]] The fact that the paper does not come with substantial theoretical contributions/justification still stands out.[[PDI-NEG,MET-NEG], [EMP-NEG]] The authors present a variant of the adversarial feature learning (AFL) approach by Edwards Storkey.[[RWK-NEU]] AFL aims to find a data representation that allows to construct a predictive model for target variable Y, and at the same time prevents to build a predictor for sensitive variable S.[[RWK-NEU]] The key idea is to solve a minimax problem where the log-likelihood of a model predicting Y is maximized, and the log-likelihood of an adversarial model predicting S is minimized. [[RWK-NEU]] The authors suggest the use of multiple adversarial models, which can be interpreted as using an ensemble model instead of a single model.[[MET-NEU]] The way the log-likelihoods of the multiple adversarial models are aggregated does not yield a probability distribution as stated in Eq. 2.[[EXP-NEG,MET-NEU], [EMP-NEG]]] While there is no requirement to have a distribution here—a simple loss term is sufficient—the scale of this term differs compared to calibrated log-likelihoods coming from a single adversary.[[EXP-NEU,MET-NEU], [EMP-NEU]] Hence, lambda in Eq. 3 may need to be chosen differently depending on the adversarial model. Without tuning lambda for each method, the empirical experiments seem unfair.[[EXP-NEU,MET-NEU], [EMP-NEG]] This may also explain why, for example, the baseline method with one adversary effectively fails for Opp-L.[[RWK-NEU]] A better comparison would be to plot the performance of the predictor of S against the performance of Y for varying lambdas. The area under this curve allows much better to compare the various methods.[[EXP-NEU, MET-NEU], [CMP-NEU]]There are little theoretical contributions. Basically, instead of a single adversarial model—e.g., a single-layer NN or a multi-layer NN—the authors propose to train multiple adversarial models on different views of the data.[[MET-NEU], [EMP-NEG]] An alternative interpretation is to use an ensemble learner where each learner is trained on a different (overlapping) feature set.[[MET-NEU]] Though, there is no theoretical justification why ensemble learning is expected to better trade-off model capacity and robustness against an adversary.[[MET-NEG], [EMP-NEG]] Tuning the architecture of the single multi-layer NN adversary might be as good?[[MET-NEU], [EMP-NEU]]In short, in the current experiments, the trade-off of the predictive performance and the effectiveness of obtaining anonymized representations effectively differs between the compared methods. This renders the comparison unfair.[[RWK-NEU, EXP-NEG], [CMP-NEG]] Given that there is also no theoretical argument why an ensemble approach is expected to perform better,[[MET-NEG], [EMP-NEG]] I recommend to reject the paper.[[OAL-NEG], [REC-NEG]]. https://openreview.net/forum?id=ByuP8yZRb.

- **Layer 3: Review Statement Purpose**

- **Layer 4: Review Statement Significance**

We also record the sentiment of the reviewer for layer 1 and 2. The sentiment of reviewers in peer review texts are a good indicator of the reviewer's implicit view towards the work [36]. Kindly see our example annotations as per Tables 2 and 3 for Reviews R1, R2, R3 above.

> ## Box 4: Annotated Review-R1 (Layer 3 and 4)
>
> There are hardly any details given on the corpus collection, the annotation method is flawed and the classification process is not really described. [[DFT,CRT], [MAJ]] What labels are you using? [[QSN], [MIN]] What are "M & S" and "H & M"?[[QSN], [MIN]] Is the data available to the community? [[QSN], [GEN]] Even the references are inadequate. [[CRT], [MAJ]] In conclusion,this manuscript was likely submitted by a student before their supervisor had the opportunity to approve it. [[CRT], [MAJ]] I suggest the authors revise the paper thoroughly and seek the assistance of senior colleagues before considering a re-submission.[[SUG,FBK], [MAJ]].

Historically peer review data are not publicly available due to its sensitivity to the authors and publisher proprietary reasons. However, with the rise of open and transparent peer review models [37], certain scholarly venues are undertaking efforts to make the peer reviews public (e.g., journals like PLOS ONE, F1000 Research, etc.) to foster transparency and trust in the system as well as support relevant research. We obtain our peer review data from the Open Review platform where the paper, peer review texts, decisions, recommendations for ICLR conference papers are publicly available. The **contributions** of the current work are:

1. We present a multi-layered annotated dataset of 1199 peer reviews, labeled for four objectives, *viz.* paper section correspondence, paper aspect, statement purpose, and statement significance. We also annotate the corresponding sentiment label for objectives 1 and 2.

2. We present five new NLP tasks on peer review texts: Review Paper Section Correspondence, Paper Aspect Category Detection, Review Statement Role Prediction, Review Statement Significance Detection, and Meta-Review Generation. We also conduct several experiments and present the baselines to the community to investigate further.

To the best of our knowledge, such a benchmark NLP resource for peer review analysis steered towards *peer review quality* is not available. We are also expanding on our annotated data and would be releasing subsequent versions of the corpus. We anticipate that our Peer Review Analyze dataset would be a valuable resource for relevant research to the NLP, Information Retrieval (IR), Meta Science, Scholarly Communications, and Peer Review communities.

## 2 Related work

Peer Review Quality has been an important research topic in the Meta Science community since the inception of the Peer Review Congress in 1989 https://peerreviewcongress.org. Here in this section, we discuss some specific studies dedicated to peer review quality. Authors in [38] studied a randomized control trial to see the effect of masking author identity to improve peer review quality. Schroter et al. [39] studied the effects of training on the quality of peer reviews. Jeffersson et al. [40] developed approaches to measure the quality of editorial peer reviews. The Review Quality Instrument (RQI) was proposed by Van Rooyen et al. [10] to assess peer reviews of manuscripts. Shattell et al. [41] studied the author's and editor's perspective on peer review quality in three scholarly nursing journals. Van Rooyen [42] proposed an evaluation framework for peer review quality. A randomized control trial to see how mentoring new peer reviewers to improve review quality was done by Houry et al. [43]. A systematic

review and meta-analysis on the impact of interventions to improve the quality of peer reviews of biomedical journals were conducted by Bruce et al. [44]. Enserink [45] explored the dubious connection between peer review and quality. D'Andrea and O'Dwyer [46] argued if editors can save peer reviews from peer reviewers. Rennie [9] proposed directions to make the peer review process scientific. Callaham et al. [47] investigated the reliability of the editor's subjective quality ratings of peer review of manuscripts. Sizo et al. [48] provides an overview of assessing the quality of peer review reports of scientific articles. However, none of these works attempted to estimate the peer review quality based on linguistic aspects automatically.

The goal for developing this dataset is to analyze and understand the reviewers' thrust over certain sections and aspects of the paper and then use those insights to investigate further the quality of peer reviews and other downstream challenges associated with the peer review system. Our research differs from the earlier works because we attempt a computational perspective (NLP/ML) to the problem. Kang et al. [49] came up with the PeerRead dataset on peer reviews. The CiteTracked dataset [50] is another dataset of peer reviews and citation statistics covering scientific papers from the machine learning community and spanning six years. However, both these datasets are a collection of peer reviews from the open review platform and are not annotated with the objectives we investigate. Ghosal et al. [36] investigated a deep network to predict the recommendation scores of the reviews and fate of the paper from the paper, reviews, and sentiment of the peer reviews. Wang and Wan [51] investigated sentiment analysis on peer review texts. Sculley et al. [52] proposed a rubric to hold reviewers to an objective standard for review quality. Superchi et al. [53] presents a comprehensive survey of criteria tools used to assess the quality of peer review reports in the biomedical domain. Wicherts [1] proposed that the peer-review process's transparency may be seen as an indicator of the quality of peer-review and developed and validated a tool enabling different stakeholders to assess the transparency of the peer-review process. Xiong et al. [54] proposed NLP techniques to provide intelligent support to peer review systems to automatically assess students' reviewing performance with respect to problem localization and solution. Ramachandran et al. [55] used metrics like review content type, review relevance, review's coverage of a submission, review tone, review volume and review plagiarism to do metareview or review of reviews. Quality in peer review is an active area of research within the peer review, meta-research, and scholarly communication communities (especially in the biomedical domain) with focused events like Peer Review Week https://peerreviewweek.wordpress.com, Peer Review Congress, and COST Action PEERE New Frontiers of Peer Review consortium https://www.peere.org/.

## 3 Dataset description: Peer review analyze

As we mentioned, we obtain our peer review data from the open review platform https://openreview.net/about. We annotate 1199 reviews from the 2018 edition of ICLR (and are continuing to expand on the annotation). Table 1 shows the dataset statistics. We collect our data from both the Accepted (ACC), Rejected (REJ), and Withdrawn (WDR) papers of ICLR 2018

**Table 1. Peer review analyze data statistics, L→# of sentences, Std→Standard deviation.**

| Category | # Papers | # Reviews | Min L | Max L | Avg L | Std | # Sentences |
|:---:|:---:|:---:|:---:|:---:|:---:|:---:|:---:|
| ACC | 184 | 555 | 2 | 82 | ∼14 | 8.52 | 7736 |
| REJ | 192 | 578 | 1 | 60 | ∼14 | 8.58 | 8190 |
| WDR | 22 | 66 | 2 | 44 | ∼16 | 9.88 | 1050 |
| Total | 398 | 1199 | - | - | - | - | 16976 |

to do a fair study of the peer reviews. We can see that there is not much difference in average review length between the three classes of papers.

## 3.1 OpenReview platform

OpenReview aims to promote openness in scientific communication, particularly the peer review process, by providing a flexible cloud-based web interface and underlying database API enabling the following: Open Peer Review, Open Publishing, Open Access, Open Discussion, Open Directory, Open Recommendations, Open API, and Open Source. ICLR follows the openreview model where the authors submit their papers to the platform, and the paper is open to review by the community until a specified date. The community can review/comment on the paper either anonymously or non-anonymously, and the reviews/comments are visible to all. After the specified deadline, the official ICLR appointed reviewers would review the article, and while doing so, they may consult the feedback that the paper received from the community. The authors can answer the community and reviewer queries/comments all through the review process. Finally, the conference chairs would decide upon the fate of the submission considering the official reviews, recommendation scores, response of the authors, and sometimes the public feedback as well. The entire process is open and visible to the public. The openreview platform provides a dedicated set of API's to crawl for the submission, its official reviews, public comments, official recommendation scores, and the final decision.

## 3.2 Annotation layers, schema, and guidelines

Keeping our investigation objectives in mind, we annotate the peer review texts in four layers, *viz. Layer 1: Review-Paper Section Correspondence*, *Layer 2: Review-Paper Aspect*, *Layer 3: Review-Statement Purpose* and *Layer 4: Review-Statement Significance*. Tables 2 and 3 show our annotation labels and guidelines with examples for the four different layers respectively.

**3.2.1 Layer 1: Review-paper section correspondence.**   The review-paper section layer identifies the section of the paper on which the review-statement is commenting. We describe the labels in Table 2 with examples. The labels are simple and obvious if one is familiar with Machine Learning papers. Not all the labels are very prominent in the review texts (e.g., Abstract (ABS), Introduction (INT), Future Work (FWK)) as the reviewers do not generally comment on certain sections unless there are some explicit issues to highlight. We find that some labels like Methodology (MET), Experiments (EXP), Results (RES), Analysis (ANA) are very inter-linked and sometimes can be hard to distinguish in the context of a review-text.

**3.2.2 Layer 2: Review-paper aspect category.**   The review-paper aspect layer identifies the aspect of the paper that the review-statement addresses. Our aspect-labels [49] (Table 2) are intuitive for an empirical domain like Machine Learning (ML). For definitions of each aspect-label, please follow the appendix section in [49]. Even though ICLR review criteria do not explicitly command these aspects, an ideal review is expected to more or less address those. Note that it may so happen that a review sentence may not conform to any of the aspects which are prescribed in the ACL 2016 reviewing guidelines [49]. We ask our annotators to leave those instances.

**3.2.3 Layer 3: Review-statement purpose.**   The review-statement purpose layer identifies the purpose or the role of the peer review statement within the peer review. It also uncovers the intent of the reviewer while writing the peer review statement. Table 3 details the various statement purposes with examples. We agree that these are not the exhaustive set of review purposes, reflecting the reviewer's intent, and there could be more. Also, some labels are related in scope, e.g., Deficit and Criticism. Reviewers usually criticize when they highlight a deficit. However, in an ideal, constructive review, the reviewer may point a deficit with a

**Table 2. Representative examples of review texts for different labels in Review-Paper Section Correspondence Layer (Layer 1) and review-paper aspect layer (Layer 2).**

| Layer 1 | | |
|---|---|---|
| **Label** | **Label Description** | **Example** |
| **Abstract (ABS)** | If the reviewer is explicitly commenting on the Abstract of the paper. | *The title and abstract are not very reflective of the content of the text.* |
| **Introduction (INT)** | If the reviewer is explicitly commenting on the Introduction of the paper or provides a general summary at the beginning of the review. | *This paper introduces a technique for program synthesis involving a restricted grammar of the problems that is beam-searched using an attentional encoder-decoder network.* |
| **Related Works (RWK)** | If the reviewer is talking explicitly on the Literature Section or comments on some related research. | *The general idea of multi-scale generation is not new, and has been investigated for instance in LapGAN (Denton et al., ICLR 2015) or StackGAN (Zhang et al., ICCV2017, arxiv 2017)* |
| **Problem Definition/Idea (PDI)** | Review statement that comments on the problem being investigated or the main scientific idea in the paper. | *The idea behind the paper is novel: translating language modeling into a matrix factorization problem is new as far as I know.* |
| **Data/Datasets (DAT)** | Any statement on the data/datasets/corpus used in the concerned work. | *Also, more monolingual experiments could have been conducted with state-of-the-art neural paraphrasing models on WikiQA and Quora datasets.* |
| **Methodology (MET)** | Review comments on the methods, the approach described in the paper, on details on how the problem has been addressed? | *By calculating the outer-loop policy gradient with respect to expectations of the trajectories sampled from $T_i$, and the trajectories sampled from $T_{i+1}$ using the locally optimal inner-loop policy, the approach learns updates that are optimal with respect to the Markovian transitions between the pairs of consecutive tasks.* |
| **Experiments (EXP)** | Review comments on the experimental section, parameter/hyperparameter details, training/testing configuration, what has been done, etc. | *The experiment section is thorough (it is written clearly, and all the experiments are described in a lot of details) and supports reproducibility.* |
| **Results (RES)** | Comments on the results, the outcome of the experiment | *The paper only shows results on image generation from random noise.* |
| **Tables & Figures (TNF)** | Comments explicitly specifying the tables and figures within the paper | *The figures in this paper depend excessively and unnecessarily on color.* |
| **Analysis (ANA)** | Comments on analysis of results, studies on the outcome | *Once again, it would be interesting for the paper to study why they achieve robustness to noise while the effect does not hold in general.* |
| **Future Work (FWK)** | Comments on the future of the work, impact on the community, etc. | *This will give the hardware community a clear vision of how such methods may be implemented both in data centers as well as on end portable devices.* |
| **Overall (OAL)** | We keep the Overall label for those review comments which are not confined to a certain section of the paper and are a comment on the overall work in general, sometimes overlaps with the Introduction label. | *Despite these questions, though, this paper is a nice addition to deep learning applications on software data, and I believe it should be accepted.* |
| **Bibliography (BIB)** | Any straightforward comments on the references or on the bibliography section of the paper | *I found some references are incomplete and should be expanded.* |
| **External (EXT)** | To justify their point, sometimes the reviewer brings external knowledge from their expertise in the review, which cannot be classified into the other section-labels. We mark those with EXT. | *Mujoco + AI Gym should make this really easy to do (for reference, I have no relationship with OpenAI)* |
| Layer 2 | | |
| **Appropriateness (APR)** | If the reviewer comments on the scope of the article to the conference or the standard/suitability of the article to the venue. | *This time the paper does not deserves to be published under ICLR* |
| **Originality or Novelty (NOV)** | Review-comments on novelty or originality of the submission. | *The main novelty in this paper is the choice of models to be used by speaker and listener, which are based on LSTMs and convolutional neural networks.* |
| **Significance or Impact (IMP)** | If the reviewer comments on the significance of the work described (e.g., inspire new ideas, insights which could be impactful to the community). | *Contains more ideas or results than most publications in this conference; goes the extra mile.* |
| **Meaningful Comparison (CMP)** | If the reviewer comments whether the work is compared against earlier approaches or where do the work stands against existing literature if the references are adequate. | *Bibliography and comparison are somewhat helpful, but it could be hard for a reader to determine exactly how this work relates to the previous work.* |
| **Presentation & Formatting (PNF)** | Review-comments on presentation and formatting aspects of the paper. | *Section 2, paragraph 3: "is given in Fig 1" -> "is given in Algorithm 1"* |

*(Continued)*

**Table 2.** (Continued)

| | | |
|---|---|---|
| **Recommendation (REC)** | Overall recommendation of the reviewer on the article for inclusion/exclusion from the proceedings. | *Overall this seems to represent a strong step forward in improving the training of GANs, and I strongly recommend this paper for publication.* |
| **Empirical & Theoretical Soundness (EMP)** | If the reviewer comments on the soundness of the approach or if the approach is well-chosen (e.g., if the arguments in the paper are cogent and well-supported). | *The empirical results appear promising, and in particular in comparison with Q-Prop, which fits Q-function using off-policy TD learning.* |
| **Substance (SUB)** | If the reviewer comments on the volume of work done and have enough substance to warrant publication or if the paper would benefit from more ideas and results. | *I wish the paper had explored a wider variety of dataset tasks and models to better show how well this claim generalizes, better situated the practical benefits of the approach (how much wall clock time is actually saved?* |
| **Clarity (CLA)** | If the reviewer comments about the writing and if the paper is well-structured or not, whether the contributions come out clear. | *It was easy to read the paper and understand it. The quality of the writing is high, and the contributions are significant* |

**Table 3. Label descriptions and representative examples of review texts in review-statement purpose layer (Layer 3) and review-statement significance layer (Layer 4).**

| | Layer 3 | |
|---|---|---|
| **Label** | Reviewer Intent/Label Descriptions | Example |
| **Summary (SMY)** | Provides a summary of the work reflecting his/her understanding of the work, usually at the beginning of the review. | *The paper proposes a novel approach on estimating the parameters of Mean field games (MFG).* |
| **Suggestion (SUG)** | Provides suggestions to the author to improvise or to include additional details for clarity, such as evidence, artifacts, etc. | *If, for example, the authors would have demonstrated all 8-bit training on all datasets with little performance degradation, this would seem much more useful.* |
| **Deficit (DFT)** | Highlights the major/minor flaws/shortcomings in the paper, complementing the work. Usually, the reviewer appears confident in their claim. | *In the current version, the paper does not explain the HDDA formalism and learning algorithm, which is a main building block in the proposed system (as it provides the density score used for adversarial example detection.)* |
| **Appreciation (APC)** | Applauds the author about their work highlighting positive aspects or specific sections of the paper. | *As adversarial training is an important topic for deep learning; I feel this work may lead to promising principled ways for adversarial training.* |
| **Discussion (DIS)** | Statements where the reviewer is engaging in simple explanations, providing additional insights, etc. Usually neutral in polarity. | *Practically, this leads to learning a reward function from demonstrations using a maximum likelihood approach, where the reward is represented using a deep neural network, and the policy is learned through an actor-critic algorithm, based on gradient descent with respect to the policy parameters.* |
| **Question (QSN)** | Reviewer is explicitly posing a question to the author (ending with a question mark), sometimes ask for further explanation, sometimes can highlight a deficit and bear negative polarity. | *Can authors provide some justifications of such different choices of activation functions?* |
| **Criticism (CRT)** | The reviewer is critical of the work, usually highlights a deficit, and bears explicit negative sentiment. | *The motivation for detecting adversarial examples is not stated clearly enough, and I am not sure how that correlates with the findings.* |
| **Feedback (FBK)** | The reviewer clearly brings out their view towards the work, usually leading to acceptance/rejection statements. | *In overall, this paper is an accept since it shows good performance on standard problems and invents some nice tricks to implement NN in hardware, for \*both\* training and inference.* |
| | Layer 4 | |
| **Major Comment (MAJ)** | A strong statement by the reviewer highlights their opinionated view on a major aspect/section or the entire paper (usually strength or weakness). It could highlight a critical flaw (empirical/theoretical soundness) that could not be rectified easily by the author or could be an appreciation of the work's novelty. The editor/chair should consider a major comment in their final decision-making or while writing the meta-review. | *According to the authors, "relevance" is one of the three criteria that characterize novelty detection: the document for which novelty is to be determined should be relevant to source documents; however, the proposed neural network architecture does not capture "relevance".* |
| **Minor Comment (MIN)** | Comments, which would not play a decisive role, are usually on presentation and formatting aspects, missing references, etc. and which could be quickly addressed by the author with less effort. | *There are only one mostly minor issues with the algorithm development, and the experiments need to be more polished.* |
| **General Comment (GEN)** | Are regular comments on the paper could not be classified into the above two categories. Usually are discussions and non-opinionated. | *The related work section is entirely focused on deep learning, while the experiment section is dedicated to sentiment analysis.* |

suggestive tone or ask questions. We keep the annotation multi-label to address such cases. We intend to merge some labels later to avoid ambiguity. An ideal review usually begins with the paper's summary, reflecting the reviewer's understanding of the paper concepts. The other components of the review are usually the reviewer's opinionated view towards the work, which may induce discussions, highlight merits-demerits, ask the authors for better understanding (in the rebuttal period), etc. Finally, in an ideal scenario, the reviewer clearly conveys their judgment towards the paper's fate and takes a stand.

**3.2.4 Layer 4: Review-statement significance.** The review-statement significance layer identifies the relative importance of the review statement within the peer review. The significance layer has the following labels: Major Comment (MAJ), Minor Comment (MIN), General Comment (GEN) (See Table 3). The purpose of this layer is to identify the reviewer's crucial statements, which would aid the editor/area chair to make the final decision. The crucial statements can form a part of the meta-review that the editor/area chair would write.

Kindly refer to our annotation illustration in reviews R1, R2, R3 in Section 1 to see how our annotated data looks like.

**3.2.5 Additional guidelines.** We provide certain *additional guidelines* to our annotators:

1. Perform sentence-wise annotation for each review document for both the layers. However, we allow our annotators to select the text-segment to label (but advise to do sentence-level annotations in most cases).

2. Consider sentence-context when a single review sentence does not make sense if considered in isolation. e.g., *What was the need for this?* is not clear if we do not consider the preceding context.

3. Consider **multi-label annotations** if multiple aspects or sections of the paper are in the selected text-segment.

4. Make phrase-level annotation (instead of sentence-level) if the selected text-segment (a portion of the review sentence) addresses multiple paper aspects/sections with varying sentiment polarity.

5. Put the confusing instances in the CANNOT DECIDE category. The primary investigating team would discuss and decide on the label of those instances and leave out the ambiguous instances from the annotation process.

6. Record the sentiment (POSitive, NEGative, NEUtral) associated with each label in Layer 1 and Layer 2.

## 3.3 Data procurement and pre-processing

We use the openreview.net API's https://github.com/openreview to crawl the official reviews of ICLR 2018 with the associated metadata (recommendation scores, final decision). We crawl and store each peer review for a given paper in a.txt file and rename it as *ICLR2018-Byxc4defh-R3.txt*, which signifies the third official review of the paper with id *#Byxc4defh* from ICLR 2018. We also design an interface, annotate these text files for the four different layers (c.f. Section 2.2), and record the annotations in a separate text file *ICLR2018-Byxc4defh-R3-annotated. txt*. Although the exact review text is available with the dataset, we perform certain pre-processing for our experiments: stop-word removal, stemming and removing irrelevant characters (like white spaces, newline characters, underscores), upper case to lower case conversions, and lemmatization.

### 3.4 Label distribution and analysis

Fig 1 shows the distribution of the labels across the ACC and REJ category for the four layers.

For layer 1, we could clearly see that the label distribution is relatively similar for both the paper categories (ACC and REJ). Methodology and Experiments are the sections of prime attention to the reviewers. Reviewers tend to comment more on the Methodology followed by Experiments, Related Work, Problem Definition/Idea, Results, Overall paper, Datasets, Introduction, Analysis, and Tables/Figures, respectively. Bibliography, Future Work, External Information, Abstract receive significantly less percentage of comments from the reviewers. Since ICLR is a core machine learning conference, this kind of distribution is not surprising. For layer 2, the primary aspects that the reviewers focus on are Empirical/Theoretical Soundness followed by Meaningful Comparisons. The other elements that receive the reviewer's attention in decreasing order are Substance, Clarity, Presentation and Formatting, Impact, Novelty. Recommendation and Appropriateness scores lower presence.

Fig 1(e)–1(h) also shows the distribution of the labels across the ACC and REJ categories for the third and fourth layers. For layer 3 (purpose), the label distribution is fairly similar for both the categories. Criticism, Summary, Discussion, Appreciation are the major categories of the review statements. Peer reviews are meant to be a critical evaluation of the work, which explains why a good portion of the review comments falls into this category. Quite expected that the share of critical comments for rejected papers is more than that of accepted papers. Similarly, the presence of appreciation statements is more in accepted paper reviews than their rejected counterparts. Usually, a review begins with the paper's summary, and hence we see a good portion of summary comments for both the accepted and rejected articles.

Reviewers also discuss their understanding and bring their knowledge, insights into the review. Usually, discussion statements are objective and are of neutral polarity. As ICLR follows an open review scheme, reviewers can pose questions to the authors who can suitably respond with their clarifications. We see nearly 10% of comments are questions. Questions can also highlight some deficits in the work and bear a negative polarity. Deficit comments hold 8.65% in rejected paper reviews and 6.95% in accepted paper reviews, which is understandable as rejected papers would have more shortcomings than the accepted ones. Please note that these reviews are for the submitted versions of the papers. The authors would address the weaknesses of their camera-ready versions of accepted papers. Suggestive comments find comparatively more importance in accepted (8.54%) than rejected (7.97%) paper reviews. We see reviewers rarely make their stand explicit about their recommendation, and hence we see a lesser proportion of feedback comments for both accepted (1.74%) and rejected (1.43%) paper reviews. Generally, the reviewers use the recommendation score field in the review form to put their views on the paper's fate.

Layer 4 distributions show the share of major, minor, and general comments in the dataset, which is more or less similar for the ACC and REJ category of papers. However, the proportion of general comments are comparatively less than major and minor comments for both the classes.

Table 4 and Fig 2 shows the *label distribution for each review* across the dataset for the four layers. The Max and Avg columns indicate the maximum and average presence of labels in each review. For the first layer, we see that reviewers spend most of their writings highlighting methodological concerns with the paper. Literature studies and experiments follow next in their presence within a review. There are arguments over the INT and OAL labels as our annotators interpret both of them having overlapping scopes. Generally, the authors write the paper summary and contributions in the paper's ABS and INT section. Reviewers usually start their reviews with a concise summary of the article, revealing their understanding of the context.

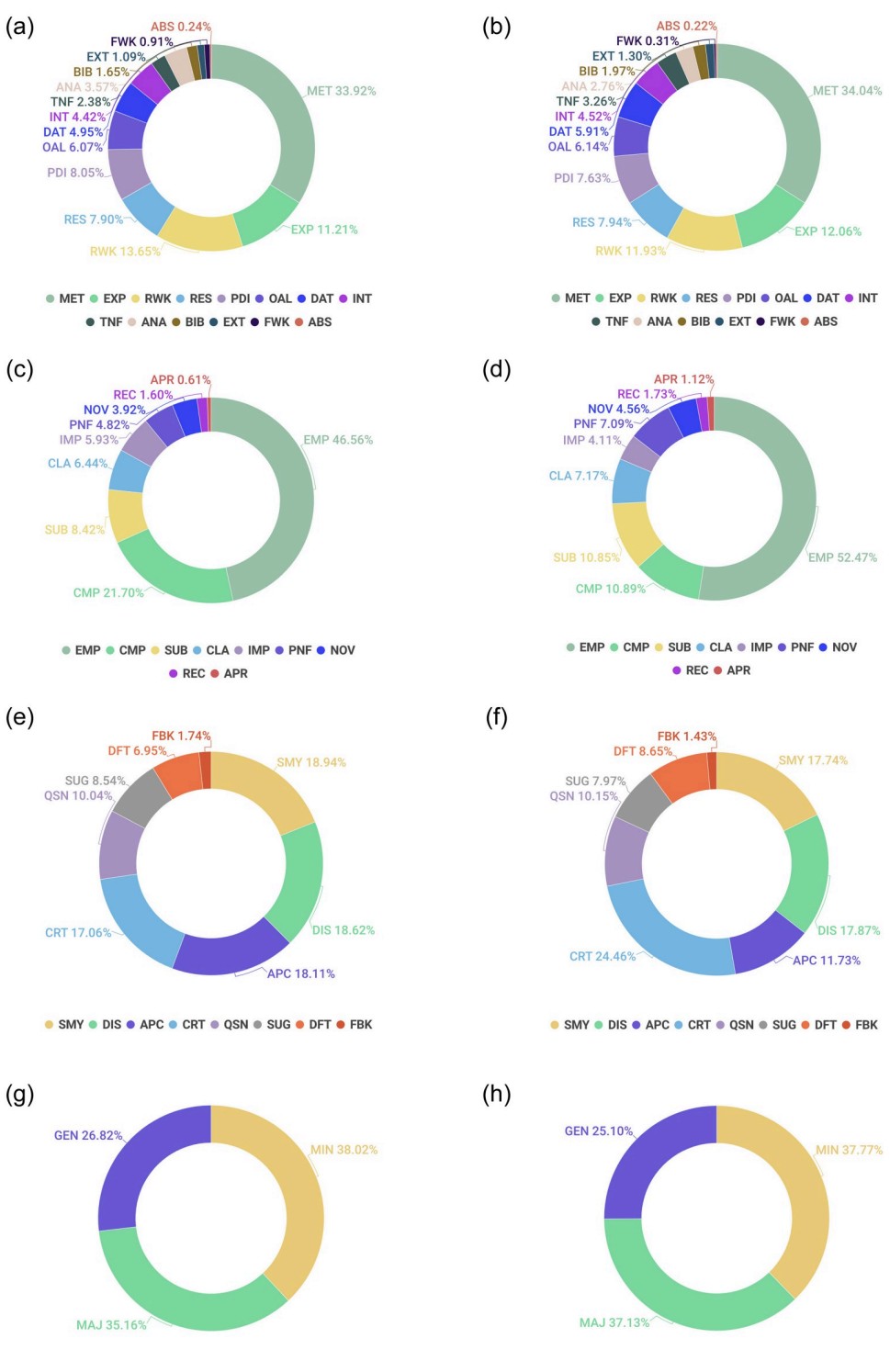

**Fig 1. Label distributions for the four layers across peer review analyze.** (a) Layer 1-ACC. (b) Layer 1-REJ. (c) Layer 2-ACC. (d) Layer 2-REJ. (e) Layer 3-ACC. (f) Layer 3-REJ. (g) Layer 4-ACC. (h) Layer 4-REJ.

**Table 4. Max and Avg occurrence of review statements in a review pertaining to the four different layers in the dataset (Min occurrence is zero for each label).**

| Layer 1 | | | Layer 2 | | | Layer 3 | | | Layer 4 | | |
|---|---|---|---|---|---|---|---|---|---|---|---|
| Labels | Max | Avg | Labels | Max | Avg | Labels | Max | Avg | Labels | Max | Avg |
| ABS | 3 | 0.05 | CLA | 11 | 0.95 | SMY | 26 | 3.34 | MAJ | 31 | 5.87 |
| INT | 9 | 1.00 | APR | 4 | 0.12 | SUG | 13 | 1.46 | MIN | 40 | 5.84 |
| RWK | 42 | 2.93 | NOV | 6 | 0.56 | DFT | 13 | 1.49 | GEN | 34 | 4.27 |
| PDI | 20 | 1.76 | SUB | 14 | 1.34 | APC | 15 | 2.53 | | | |
| DAT | 12 | 1.21 | IMP | 9 | 0.75 | DIS | 18 | 2.69 | | | |
| MET | 32 | 7.51 | CMP | 11 | 1.43 | QSN | 18 | 1.66 | | | |
| EXP | 17 | 2.61 | PNF | 14 | 0.75 | CRT | 24 | 3.38 | | | |
| RES | 15 | 1.75 | EMP | 31 | 6.93 | FBK | 4 | 0.28 | | | |
| TNF | 10 | 0.61 | REC | 3 | 0.23 | | | | | | |
| ANA | 17 | 0.71 | | | | | | | | | |
| FWK | 5 | 0.15 | | | | | | | | | |
| OVA | 10 | 1.12 | | | | | | | | | |
| BIB | 8 | 0.41 | | | | | | | | | |
| EXT | 6 | 0.27 | | | | | | | | | |

We mark the brief summary of the paper at the beginning of the review, as INT and sometimes as OAL. Overall comments can also appear in the middle or end of the review. So, to avoid confusion, if we club the two annotations, OAL comes next as reviewers often tend to make generic comments on the overall work. Problem Definition/Idea, Results, Datasets are their next concerns in a review. Obviously, these statistics are average and would vary depending on the paper and venue. Tables and Figures, Analysis, Bibliography, find lesser concerns in a review.

Table 4 also shows the *average label occurrence for paper aspects in a review*. Reviewers mostly comment on the paper's empirical and theoretical aspects in the review, which is quite understandable in the context of ICLR. Reviewers then tend to comment on the comparison with the existing literature followed by the substance or volume of works in the paper. Presentation, Impact, Clarity of writing are the aspects that receive more or less equal attention from the reviewers. Surprisingly, novelty proportion is low here since reviewers may not explicitly mention the newness of the work in their review. Still, the aspect is often implicit in their review. For the recommendation aspect, reviewers announce their overall view regarding the paper in the recommendation scores and seldom make their recommendation explicit in the review text. The appropriateness fraction is low. This is because ICLR is a widely publicized top-tier ML conference. Generally, authors do not make uninformed submissions; hence, reviewers do not have to spend time judging if the submission is *within scope* [56].

For layer 3, we could clearly see the average presence of CRT, SMY, DIS, and APC are higher for each review, signifying that reviewers follow the general structure of a review that consists of a concise summary, discussions, and pros and cons of the work.

For layer 4, the average presence of major and minor comments in a review are almost similar.

## 3.5 Interaction between the layers

Since we annotate the same text at four different layers, we want to understand how the labels in the two layers interact. Hence, we map the label co-occurrence in Figs 3 and 4 across different layers. Although a multi-labeling task, we take co-occurrence of a label in one layer (Section) with a label in the other layer (aspect) to see which section receives frequent comments

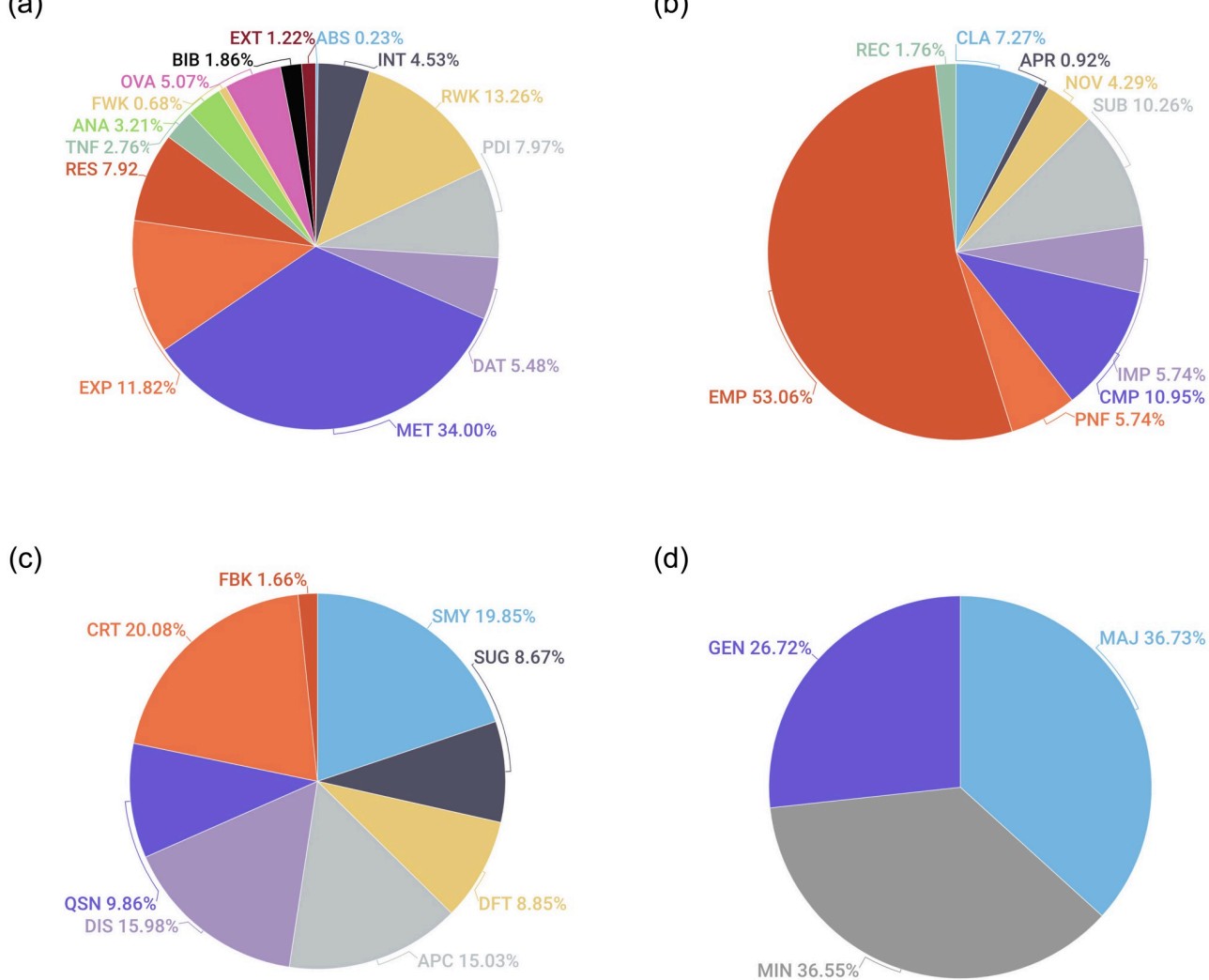

**Fig 2. Label distribution pie charts showing the relative importance of each category of statements in one single review across the four layers.** The average label occurrence for each layer in Table 4 is translated to percentage distribution in these pie charts. (a) Layer 1. (b) Layer 2. (c) Layer 3. (d) Layer 4.

on which aspect. Fig 3 shows that while reviewers comment on the Clarity aspect, they do so on the Overall paper followed by Clarity in the Methodology section. The same goes for Appropriateness; a reviewer would generally comment on the entire paper's suitability to the venue. For the Novelty aspect: Methodology, Problem Definition/Idea, and Overall are the sections on which the reviewers tend to write more. Reviewers also tend to comment on the Related Work while talking about Novelty, which could be due to comparing the current work with respect to the existing literature. Concerning the Substance aspect, reviewers mostly relate to the following sections in order: Methodology, Experiments, Related Work, Analysis, Data Description, and Results, which eventually form the nucleus of the paper. While discussing the Impact of the work on the community, reviewers mostly focus on the Methodology, Related Work, Experiments, Results, Problem Definition/Idea, and Overall paper. We also see reviewers discuss the Future Impact of the work. Quite understandable that the Meaningful Comparisons aspect relates most to the Related Work and Methodology. Also, with Experiments and

(a)

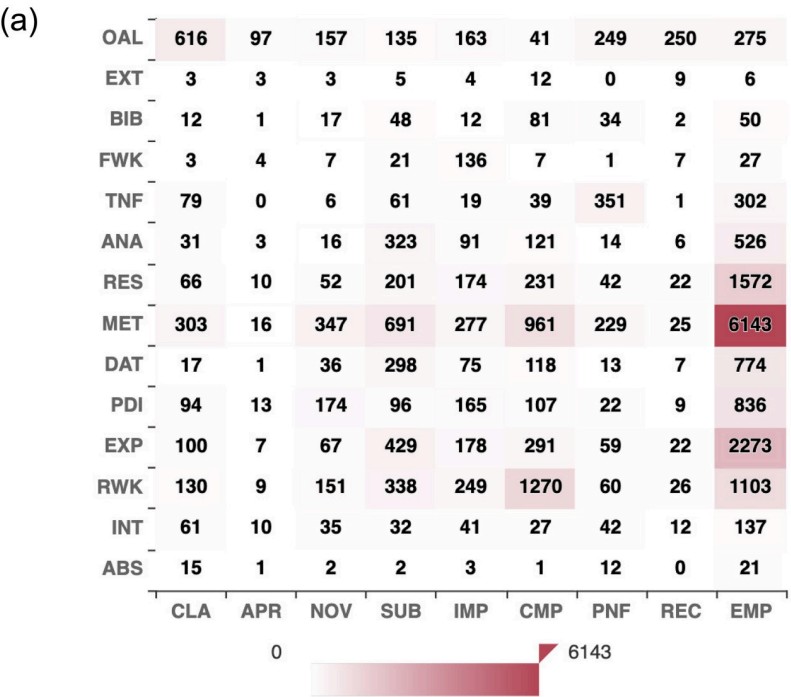

(b)

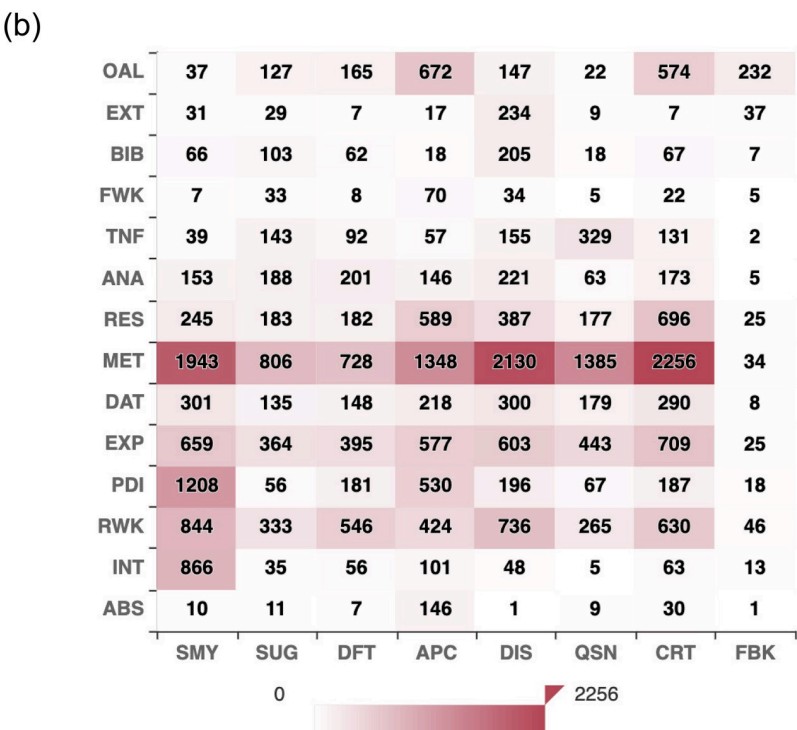

**Fig 3. Heatmaps showing the label co-occurrence between two layers highlighting inter-dependency between the layers.** (a) Layer 1 vs Layer 2. (b) Layer 1 vs Layer 3.

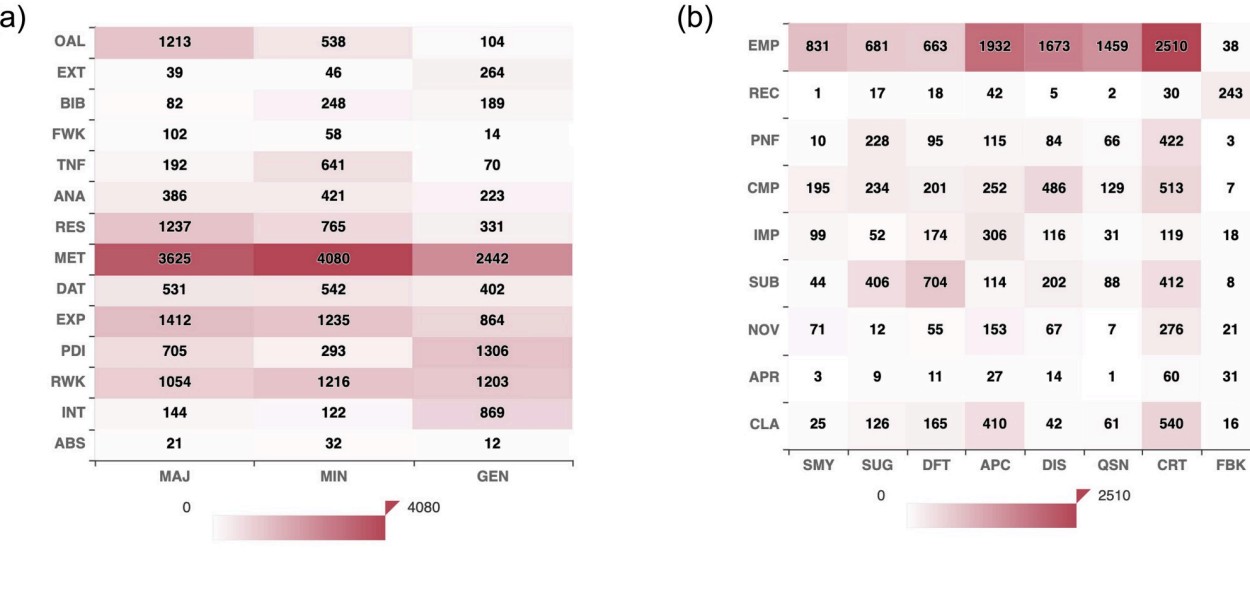

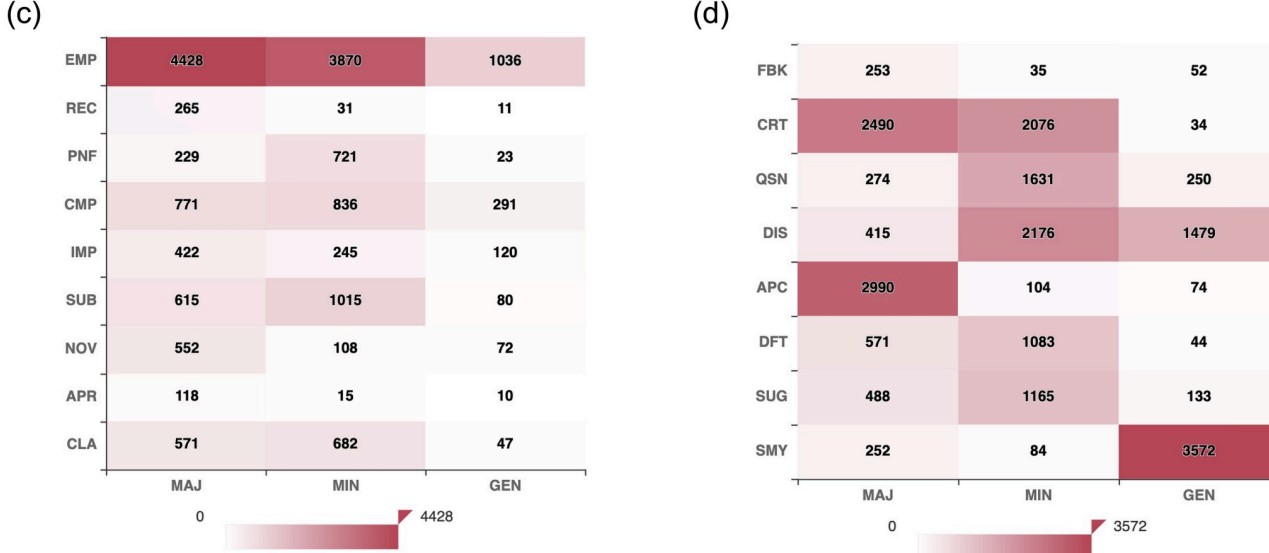

**Fig 4. Heatmaps showing the label co-occurrence between two layers highlighting inter-dependency between the layers.** (a) Layer 1 vs Layer 4. (b) Layer 2 vs Layer 3. (c) Layer 2 vs Layer 4. (d) Layer 3 vs. Layer 4.

Results. The Bibliography section also co-occurs with the Meaningful Comparison aspect as sometimes the reviewers highlight the bibliography to compare the current paper with the earlier works. Presentation and Formatting concerns are mostly on Tables and Figures, Methodology, and Overall. Reviewers usually put their Recommendation when they talk about the paper in its entirety. Finally, the most significant aspect: Empirical and Theoretical Soundness is highly judged against the Methodology section, followed by Experiments, Results, Related Work, Problem Definition, Data, and Analysis. This is expected considering the scope of ICLR as a high-profile core ML venue.

Inter-dependency heatmap between Layer 1 and Layer 3 reveals: when reviewers summarize their understanding, they mostly focus on the Methodology and the Problem Definition. When reviewers tend to Suggest, they do so mostly on Methodology, Experiments, and Related

Works. Reviewers primarily highlight Deficits on Methodology, Related Works, Experiments, and Analysis. Similarly, Appreciation and Criticism mostly go with Methodology, Experiments, Results, Related Works, and Overall Work. Reviewers also appreciate the Problem Definition, whereas critics on Problem Definition is comparatively low. This could be due to that authors are aware of the stature of ICLR in ML domain h5-index of ICLR is 203, which is highest in AI, according to Google Scholar; ICLR launched in 2013 and generally consider to submit their very best ideas there. So, usually, the problems are very interesting. Reviewers spend most of their Discussions on the Methodology followed by Related Works, Experiments, and Results, which was evident from our analysis in Table 4 as well. Questions are mostly on Methodology followed by Experiments, Tables and Figures, and Related Work. Quite obvious that reviewers give Feedback on the Overall paper.

Inter-dependency heatmap between Layer 1 and Layer 4 (Fig 4a) reveals that most of the Major, Minor, General comments of the reviewers are on the Methodology, Related Works, Results, Experiment sections, and also on the Overall paper.

Inter-dependency heatmap between Layer 2 and Layer 3 (Fig 4b) shows that all significant labels of Layer 3 mostly go with Empirical and Theoretical Soundness label in Layer 2. Certain exceptions: Reviewers highlight Deficit in the Substance of the paper, and while giving Feedback on the overall paper, reviewers are actually making their Recommendations clear. While making Suggestions, reviewers comment on Substance (like how to improvise on the volume of contribution to warrant a publication at ICLR), Presentation and Writing issues, and Comparison with earlier works. Along with Empirical and Theoretical soundness, reviewers also appreciate the Clarity in writing and composition. Reviewers also tend to Discuss how the current work stands in comparison to the existing literature. As we said, reviews are meant to be critical scrutiny of the paper; hence we see reviewers Critique about almost all important aspects.

Quite understandable that the significance labels in Layer 4 are mostly on EMP in Layer 2 (Fig 4c), which was also the case with Methodology in Layer 2 (MET and EMP are highly correlated). Reviewers put their major comments on certain aspects like Meaningful Comparison, Impact, Substance, Novelty, and Clarity. Apart from these, we see a good number of Minor comments on the Presentation and Formatting aspect, the reason for which can be easily perceived.

Inter-dependency between Layer 3 and Layer 4 (Fig 4d) reveals some interesting observations: Major comments highlight Criticism and Appreciation, which are decisive for the paper. Minor comments mostly Discuss some points, put Critics on the work, ask Questions to the authors, point out Deficits, and also provide Suggestions. General comments constitute a Summary of the work under review and general Discussions (non-opinionated) in the review.

## 3.6 Sentiment distribution

Fig 5 shows the distribution of sentiment information across the different labels in layer 1 and layer 2 for the accepted and rejected paper reviews. We see that the NEUtral label is dominant for the majority of the sections. This signifies that most review texts are objective in nature and that the reviewers discuss the facts (non-opinionated) on the sections. But there are a good number of POSitive and NEGative instances that reveal the reviewer's attitude towards the work. As peer review tends to be critical, we see a good presence of NEGative comments in accepted paper reviews as well. Also, acceptance or rejection of a paper does not depend on one single review, and reviews may be critical yet suggesting acceptance.

The EMP, EXP, OAL, RES, NOV, REC, IMP labels have higher positive instances for the accepted paper reviews, while higher negative instances for the rejected paper reviews, which

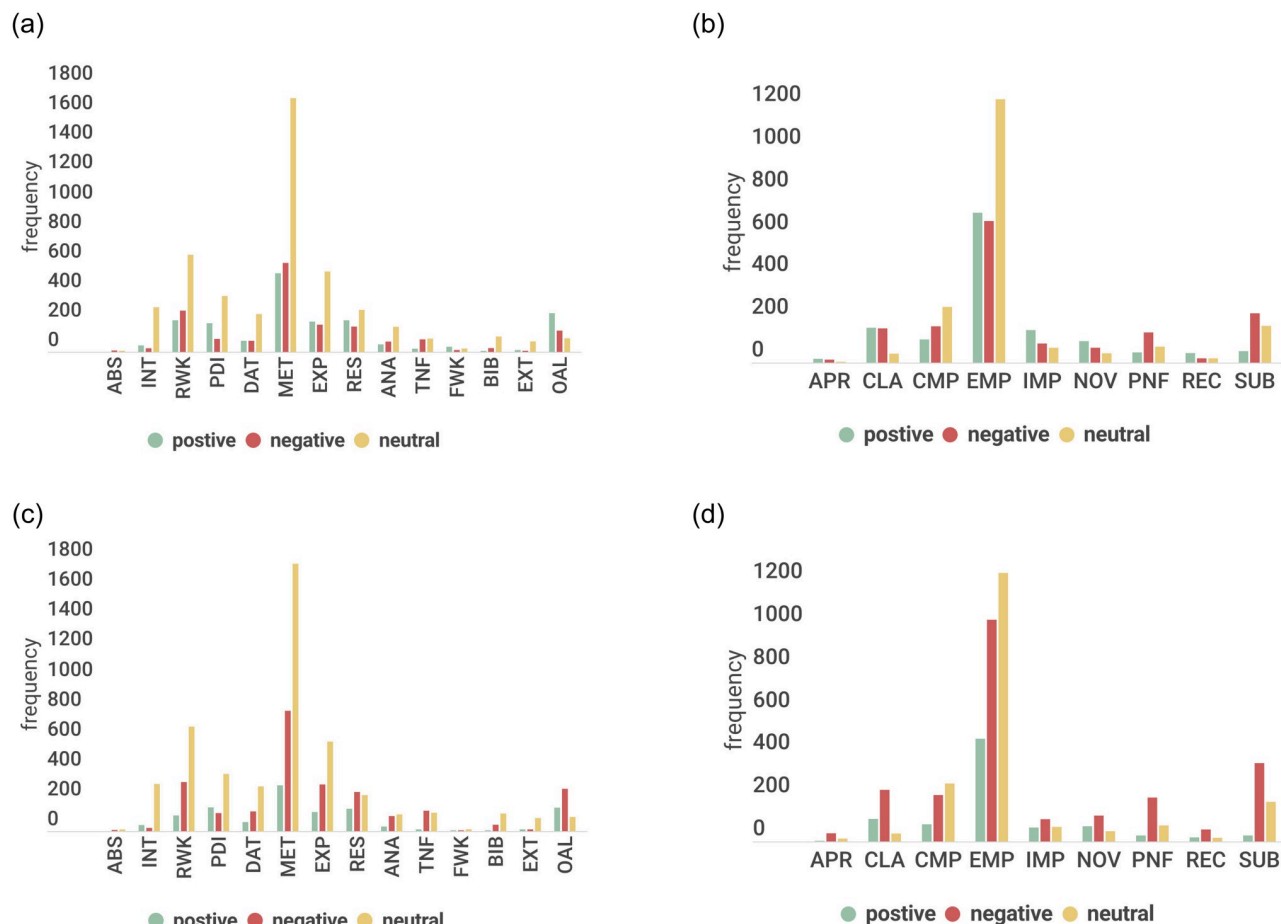

**Fig 5. Sentiment distribution for labels across Layer 1 and Layer 2 for ACC and REJ papers.** (a) Sentiment for Layer1-ACC. (b) Sentiment for Layer2-ACC. (c) Sentiment for Layer1-REJ. (d) Sentiment for Layer2-REJ.

are justified. These are in fact the most common sections/aspects on which the reviewers tend to invest their attention while scrutinizing the paper. Thus sentiment of the reviewers vary with their evaluation of the paper and are hence most prominent with these sections. Reviewers are more critical of the RWK, MET sections, and SUB aspect. The reason could be that these sections/aspects tend to be more critically reviewed than others as the reviewers criticise if the authors missed crucial citations in the related works, highlights deficits or questions the methodology or approaches. Also, the volume of work done or Substance in the paper attracts criticisms of the reviewers. For TNF and PNF, reviewers generally comment when they want to highlight an issue (usually negative). Reviewers tend to be NEUtral when commenting on the BIB. We see that PDI garners comparatively positive attention from the reviewers for both the accepted and rejected papers. This could be due to the fact that authors usually do not submit a subpar problem definition or idea at a high-quality conference like ICLR. CMP and CLA in writing attract more negative comments than positive for both the classes of paper reviews.

### 3.7 Data quality

We appoint three annotators for the task of verifying the data quality. Two annotators (also authors) hold a Master's degree in Computer Science and Engineering and are currently Ph.D.

**Snapshot of our 4-layer peer review annotation interface**

**Fig 6.**

students (the first author is about to graduate, the third author is in the second year) and are familiar with NLP/ML paper discourse and review structure. The data annotation forms an integral part of their thesis. The third annotator is an engineering bachelor's graduate in Computer Science and is exclusively hired full-time for the task. We intentionally employ our annotators from a technical background as they would understand the scholarly texts better. The confusion cases are resolved in team meetings comprising all the primary investigators. The entire review and the corresponding paper were made available to the annotators during labeling for better understanding. We design an easy to use interface (see Fig 6) where the annotators are required to upload the reviews and annotate the various labels via selecting the text (usually sentences) and marking via checkboxes pertaining to the two different layers. We also conduct a rigorous two-month exercise involving all the annotators on the peer review process/aspects, general machine learning paper contents and format, subjectivity, and labels of the four layers. The annotation period lasted for more than ten months. On average, it took $\sim$40 minutes to annotate one review of average length, but the time decreased as we progressed in the project. Apart from our guidelines, we instructed our annotators to read the entire review first to understand the reviewer's context and attitude better. We also measure the inter-annotator agreement on a subset of data (100 full reviews for the two layers). Considering a multi-label scenario, Inter-Annotator Agreement (IAA) Krippendorff's Alpha [57] for layer 1, layer 2, layer 3, and layer 4 are 0.86, 0.70, 0.73, and 0.84 respectively. We continue to expand the inter-annotation range to come up with a better representative IAA.

## 4 Associated tasks description

As we mention earlier, we define five *new* tasks on peer review text with this dataset. Although our tasks are motivated towards *peer review quality*, we conduct our experiments on the four

scaffolding NLP tasks. We hypothesize that these four tasks would enable us to objectively attempt a computational approach towards *peer review quality*. We invite the larger NLP, Machine Learning, and Meta Science community to explore our dataset for the main problem and several sub-problems associated with peer reviews.

### 4.1 Task 1: Review-paper section correspondence

Given a peer review text, can we identify which section(s) of the paper does it correspond to? We take the general sections of a machine learning paper, as indicated in Table 2. Not all the sections are explicit in a paper, and not all reviews are required to address all of them. The task is a multi-label classification one, as a review text can correspond to multiple sections in the paper.

### 4.2 Task 2: Aspect-category detection

Given a peer review text, can we identify the aspect(s) of the paper on which the reviewer is commenting? We consider the aspects, as shown in Table 2. Like Task 1, a review text can address multiple aspects of the paper; hence, we also view Task 2 as a multi-label classification problem.

An allied task would be: can we identify the sentiment of the reviewer encoded in the peer review text? [36]. How that aspect/section-specific sentiment would steer the final decision? Since, we record the sentiment of the reviewer for both section and aspect layer, this would be an interesting problem to pursue.

### 4.3 Task 3: Review-statement purpose detection

Given a peer review statement, can we identify the role played by that statement in the peer review? We define the following roles: *Summary, Suggestion, Deficit, Appreciation, Discussion, Question, Criticism*, and *Feedback* as per Table 3. This task would help to know the reviewer intent in the peer review.

### 4.4 Task 4: Review-statement significance detection

Given a peer review statement, can we identify the significance of the statement in the peer review? Can we determine if the statement is a *significant comment, a general discussion, or a trivial (minor) comment?*

### 4.5 Task 5: Peer review summarization

Building upon Task 2, given a peer review, can we summarize and highlight the crucial aspects of the review that would aid the area chairs to draft the meta-review? Can we go even one step further to generate the *meta-review* itself?

## 5 Evaluation

We evaluate the five tasks and provide baseline results on Peer Review Analyze. For the first four tasks, we report the results of popular feature representation schemes and also curate a set of HandCrafted Features (HCF). Finally, we provide a recent Bidirectional Encoder Representation from Transformers (BERT) [58] based baseline for sequential sentence classification. We deem Tasks 1 to 4 as the sentence classification tasks where context is crucial. Hence, sequential sentence classification is appropriate. The fifth task is close to text summarization which we evaluate with a popular unsupervised method and one supervised attention-based neural baseline. We appeal to the larger NLP/ML community to develop better evaluation

methods to address these seemingly important tasks to objectively study the peer review texts for perceived quality, purpose, and significance.

## 5.1 Baselines

Along with the traditional text feature representations, we experiment with two *state-of-the-art* deep representation schemes to identify our labels for Task 1, 2, 3, and 4. We tried with several ML classifiers, Random Forest performs the best for us.

**5.1.1 TF-IDF representation.** We take the *term frequency-inverse document frequency* representation of the peer review texts and pass it through a classifier (M1). We also, take the surrounding context (previous and next) sentences for the sentence-classification for the four layers (M2).

**5.1.2 Universal Sentence Encoder (USE).** The USE [59] encodes text into high dimensional vectors that can be used for text classification, semantic similarity, clustering and other NLP tasks. The input is a variable-length English text and the output is a 512 dimensional vector. We use the Deep Averaging Network (DAN) version of the encoder. We then pass the high dimensional semantic sentence representation to a Random Forest (RF) classifier (M3).

**5.1.3 Feature engineering.** We take some general features for both the tasks: *# of nouns, # of adjectives, exclamation sign present or not, presence of numeric values, presence of %, presence of date, # of mathematical terms, # of adverbs, length of sentence, # of verbs, # of label-specific frequent terms. We curate a dictionary of label-specific frequent words from the dataset,* presence of wh-question words, presence of interrogation character. We extract these features and pass it through a classifier (M4).

**5.1.4 BERT-based Sequential Sentence Classification (SSC).** Labelling the review sentences for the correct category is one form of sequence classification task. The contexts of the sentence (review-text) are of much importance to decide on the label for the current sentence. For this reason, we implement a latest sequential sentence classification model [60]. In this work, the authors showed that pre-trained language model, BERT [58] in particular, can be used for this task to capture contextual dependencies. We use the implementation and train the model on our dataset to identify our review-text labels for the four tasks (M5). We use a multi-layer feed-forward network on top of the [SEP] representations of each sentence to classify them to their corresponding categories. We also use the BERT + Transformer + Conditional Random Field (CRF) variant to compare our results (M6). Conditional Random Field (CRF) is popular in NLP for the sequence labelling tasks.

## 5.2 Results

We report the label-wise $F_1$ scores for Task 1 in Table 5, for Task 2 in Table 6, and for Task 3 and 4 in Table 7. Since our dataset is not balanced across all the labels, we report the label-wise micro-averaged $F_1$ scores for the different baselines. We do a 80%:20% split for training and

**Table 5. Label-wise F1 scores (micro-averaged) for Task 1, M(1-6)→Methods for evaluation, refer Table 6.**

| M | ABS | INT | RWK | PDI | DAT | MET | EXP | RES | ANA | TNF | FWK | BIB | EXT | OAL |
|----|------|------|------|------|------|------|------|------|------|------|------|------|------|------|
| M1 | 0.48 | 0.41 | 0.15 | 0.27 | 0.43 | 0.54 | 0.34 | 0.45 | 0.31 | 0.51 | 0.37 | 0.28 | 0.17 | 0.50 |
| M2 | 0.24 | 0.53 | 0.21 | 0.33 | 0.49 | 0.59 | 0.38 | 0.51 | 0.34 | 0.56 | 0.31 | 0.31 | 0.11 | 0.50 |
| M3 | 0.00 | 0.30 | 0.01 | 0.03 | 0.34 | 0.53 | 0.15 | 0.25 | 0.03 | 0.24 | 0.00 | 0.02 | 0.00 | 0.50 |
| M4 | 0.13 | 0.34 | 0.13 | 0.16 | 0.27 | 0.49 | 0.12 | 0.06 | 0.19 | 0.18 | 0.22 | 0.27 | 0.01 | 0.38 |
| M5 | **0.57** | **0.63** | **0.56** | **0.41** | **0.56** | 0.66 | **0.51** | **0.59** | **0.42** | 0.73 | 0.53 | **0.72** | **0.19** | **0.63** |
| M6 | 0.29 | 0.63 | 0.56 | 0.40 | 0.54 | **0.67** | 0.49 | 0.59 | 0.35 | **0.74** | **0.54** | 0.66 | 0.10 | 0.61 |

**Table 6. Label-wise F1 scores (micro-averaged) for Task 2.**

| Methods | CLA | APR | NOV | SUB | IMP | CMP | PNF | REC | EMP |
|---|---|---|---|---|---|---|---|---|---|
| TF-IDF + RF (M1) | 0.45 | 0.33 | 0.47 | 0.14 | 0.09 | 0.33 | 0.29 | 0.49 | 0.57 |
| TF-IDF w/context + RF (M2) | 0.62 | 0.34 | 0.59 | 0.06 | 0.10 | 0.39 | 0.32 | 0.46 | 0.64 |
| USE + RF (M3) | 0.45 | 0.05 | 0.10 | 0.01 | 0.01 | 0.19 | 0.05 | 0.26 | 0.61 |
| HCF + RF (M4) | 0.32 | 0.03 | 0.12 | 0.05 | 0.07 | 0.20 | 0.08 | 0.12 | 0.57 |
| BERT-SSC (M5) | 0.61 | 0.32 | **0.69** | **0.39** | 0.26 | 0.43 | **0.43** | **0.65** | 0.68 |
| BERT+Transformer+CRF (M6) | **0.65** | **0.55** | 0.69 | 0.38 | **0.29** | **0.44** | 0.40 | 0.62 | **0.71** |

testing instances. We perform with a range of classifiers (Support Vector Machine-SVM, Multinomial Naive Bayes, Decision Trees, Stochastic Gradient Descent (SGD) classifier), but report the best performing one which is Random Forest (RF). Although the annotation are for multiple labels, we do single-label prediction (add $n$ instances of the same text for $n$ different labels). We perform random oversampling of minority labels to address the class imbalance problem. For TF-IDF with context baseline, we concatenate the preceding and following sentences for the current sentence/review text. Clearly, we see that the BERT-based baselines outperform all the other comparing baselines for the different tasks. The BERT baselines are pretrained on large scientific domain data (using SciBERT [61] pre-trained weights) and also use the context to predict the sentence labels. We also see that TF-IDF with context performs better than USE, TF-IDF, HCF. The HCF baseline performs best to identify questions (QSN) as question statements usually contain question-mark (?), which is one of the features (Table 7). However, we assert that we would need more annotated data (for data-scarce labels), better label-specific features, and efficient sentence sequential classification (SSC) approaches. Merging some section labels may resolve ambiguities. Table 8 shows the overall task performance for the various baseline methods. While the only BERT-SSC variant performs best for

**Table 7. Label-wise F1 scores (micro-averaged) for Task 3 and Task 4.**

| Methods | Task 3 | | | | | | | | Task 4 | | |
|---|---|---|---|---|---|---|---|---|---|---|---|
| | APC | CRT | DFT | DIS | FBK | QSN | SMY | SUG | GEN | MAJ | MIN |
| TF-IDF + SVM | 0.56 | 0.42 | 0.19 | 0.28 | 0.38 | 0.36 | 0.55 | 0.42 | 0.57 | 0.58 | 0.56 |
| TF-IDF w/context + SVM | 0.54 | 0.41 | 0.19 | 0.25 | 0.42 | 0.38 | 0.58 | 0.37 | 0.59 | 0.59 | 0.61 |
| USE + SVM | 0.52 | 0.37 | 0.20 | 0.24 | 0.51 | 0.41 | 0.55 | 0.352 | 0.56 | 0.57 | 0.55 |
| HCF + SVM | 0.36 | 0.33 | 0.07 | 0.18 | 0.08 | **0.85** | 0.49 | 0.36 | 0.49 | 0.48 | 0.42 |
| BERT-SSC [60] | **0.75** | 0.61 | **0.35** | **0.51** | 0.58 | 0.79 | **0.75** | **0.61** | **0.66** | **0.67** | **0.68** |
| BERT+Transformer+CRF [60] | 0.74 | **0.62** | 0.34 | 0.46 | **0.62** | 0.83 | 0.71 | 0.57 | 0.62 | 0.66 | 0.64 |

**Table 8. Overall accuracy figures on the initial four tasks for the different baseline methods.**

| Methods | Task 1 | Task 2 | Task 3 | Task 4 |
|---|---|---|---|---|
| TF-IDF + RF | 42.97% | 44.48% | 42.10% | 57.12% |
| TF-IDF with context + RF | 48.38% | 52.3% | 41.11% | 60.12% |
| USE + RF | 38.88% | 46.39% | 39.91% | 56.08% |
| Handcrafted Features + RF | 32.53% | 39.65% | 35.71% | 43.32% |
| BERT-SSC | **58.51%** | 58.42% | **63.08%** | **65.32%** |
| BERT+Transformer+CRF-SSC | 57.61% | **59.82%** | 61.81% | 62.50% |

**Table 9. P→Precision, R→Recall, R1→ROUGE with unigram, R2→ROUGE-2 for bigram overlap, R-L→ROUGE-L for longest common subsequence.**

| Model | R1 | | | R2 | | | R-L | | |
|---|---|---|---|---|---|---|---|---|---|
| | P | R | F1 | P | R | F1 | P | R | F1 |
| Summarization on each review with MAJ comments as reference summary | | | | | | | | | |
| TextRank (Extractive) | 0.538 | 0.221 | 0.267 | 0.358 | 0.138 | 0.170 | 0.509 | 0.230 | 0.281 |
| Transformer(Abstractive) | 0.559 | 0.230 | 0.277 | 0.372 | 0.143 | 0.176 | 0.529 | 0.239 | 0.292 |
| Summarization on three reviews of each paper with the Meta Review as reference summary | | | | | | | | | |
| TextRank (Extractive) | 0.299 | 0.165 | 0.190 | 0.047 | 0.025 | 0.029 | 0.256 | 0.145 | 0.169 |
| Transformer (Abstractive) | 0.227 | 0.201 | 0.189 | 0.024 | 0.021 | 0.019 | 0.163 | 0.147 | 0.139 |

Task 1, Task 3, and Task 4, the BERT+Transformer+CRF variant performs better for Task 2 classifications.

**5.2.1 Task 5: Peer review summarization-towards meta-review generation.** For this task, we proceed with two objectives:

1. Generate a summary for each review and compare it with the annotated major comments (as gold-standard).

2. Generate a summary of the three reviews combined for a given paper and compare it with the corresponding meta-review (as the gold standard) written by the ICLR area chairs.

We experiment with an unsupervised extractive summarization technique using TextRank [62]. For abstractive summarization we make use of the encoder-decoder training with *state-of-the-art* Transformer [63] model to generate the summaries. We do an 80%-20% train-test split for the abstractive summarizer Transformer model. The baseline results for objectives 1 and 2 are in Table 9. We can see in terms of ROUGE [64] scores, both the extractive and abstractive models perform comparatively, the latter a bit better. However, the summarization performance for objective 2 is worse. This is because meta-reviews are not exactly simple summaries of the peer reviews. Generally, in meta-reviews, area chairs write their opinionated views after reading the official peer reviews. Meta-review generation would require a culmination of summarization, redundancy elimination, diversity estimation, and sentiment analysis in a natural language generation setting. *Meta-Review Generation* would be a challenging generation task and should encompass: a concise description of the submission's main content, a concise summary of the reviewer's discussion, and finally, explicit recommendation and justification on the fate of the paper https://iclr.cc/Conferences/2020/MetareviewGuide.

We want to reiterate that the current work's motivation is to present a computational perspective to the problem of *peer-review quality*. We identify certain tasks from the eye of NLP, which may steer investigations towards the main task. Our manually annotated dataset across four different layers is the first attempt towards that goal. We provide simple approaches as baselines to the community to investigate further.

## 6 Conclusion

Estimating *Peer Review Quality* is a crucial problem for the health of science and also to add force to the *gatekeeper of scientific knowledge and wisdom*. To date, there had been numerous impactful studies, experiments to improve the peer review system as well as to enhance the quality of peer reviews, mostly focused on improving the quality of peer reviews. We attempt to computationally analyze the human-generated reviews and define certain tasks that may steer automatic quality estimation of the peer reviews. With this work, we present a multi-layered dataset of annotated peer review texts and propose four novel tasks (+baselines) to

investigate peer-review quality. We also introduce the fifth task for meta-review generation and provide appropriate baselines. Our dataset is novel, multi-faceted, and we hope we can support a variety of investigations, downstream applications on peer reviews, and scholarly communications. We are currently investigating the effect of our scaffold tasks on *peer review quality* and how *sentiment* of peer reviewers can jointly infer the fate of the paper. An automated review feedback system can also aid the authors to comprehend the human reviews better, chalk out crisp action points, and thereby improvise on their manuscript.

To the best of our knowledge, such an annotated resource on peer reviews is not available. We hope that our dataset would motivate the larger NLP, Meta Science communities to take up these problems, discover sub-problems, and steer towards the larger goal of quality estimation of peer reviews. We also hope that our resource and baselines would provide an exciting testbed to the NLP/ML community for relevant research and also encourage investigations to fix our paper vetting system. An AI that would support editors/area chairs to evaluate human reviews may also persuade human reviewers to write good reviews, thereby strengthening the *holy grail of research validation*. We agree that this study is very specific to NLP/ML peer reviews. Hence as a future work, we would like to extend this study for other STEM and non-STEM fields with different ontologies, labels. It would be interesting to see how this idea to quantify review quality would generalize across other domains. Our dataset and associated codes is also available at https://github.com/Tirthankar-Ghosal/Peer-Review-Analyze-1.0.

## Acknowledgments

The first author commenced this work during his doctoral studies at Indian Institute of Technology Patna. He would like to thank Visvesvaraya PhD Scheme, Ministry of Electronics and Information Technology, Government of India for supporting his fellowship. Sandeep Kumar acknowledges the Prime Minister Research Fellowship (PMRF) program of the Govt of India for its support. The research reported in this paper is supported by Dr. Asif Ekbal's Young Faculty Research Fellowship (YFRF) Award, supported by Visvesvaraya PhD scheme for Electronics and IT, Ministry of Electronics and Information Technology (MeitY), Government of India, being implemented by Digital India Corporation (formerly Media Lab Asia).

## Author Contributions

**Conceptualization:** Tirthankar Ghosal.

**Data curation:** Tirthankar Ghosal, Prabhat Kumar Bharti.

**Investigation:** Tirthankar Ghosal.

**Methodology:** Tirthankar Ghosal.

**Project administration:** Tirthankar Ghosal, Asif Ekbal.

**Resources:** Tirthankar Ghosal, Prabhat Kumar Bharti.

**Software:** Tirthankar Ghosal, Sandeep Kumar.

**Supervision:** Tirthankar Ghosal, Asif Ekbal.

**Validation:** Tirthankar Ghosal.

**Writing – original draft:** Tirthankar Ghosal.

**Writing – review & editing:** Tirthankar Ghosal, Asif Ekbal.

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
