## [Decision Letter · Decision Letter 0]

4 Jan 2021

PONE-D-20-36706

Peer Review Analyze: A Novel Benchmark Resource for Computational Analysis of Peer Reviews

PLOS ONE

Dear Dr. Ghosal,

Thank you for submitting your manuscript to PLOS ONE. After careful consideration, we feel that it has merit but does not fully meet PLOS ONE’s publication criteria as it currently stands. Therefore, we invite you to submit a revised version of the manuscript that addresses the points raised during the review process.

We look forward to receiving your revised manuscript.

Kind regards,

Lianmeng Jiao

Academic Editor

PLOS ONE

Journal Requirements:

2.Thank you for stating the following in the Acknowledgments Section of your manuscript:

"The first author thanks Visvesvaraya Ph.D. Fellowship Award (# VISPHD-MEITY-2518) from the

Ministry of Electronics and Information Technology (MEITY), Government of India. The fourth

author is also a recipient of the Visvesvaraya Young Faculty Award and acknowledges Digital India

Corporation, MEITY for supporting this research."

3. Please upload a copy of Supporting Information S1: Dataset and Codes which you refer to in your text in line 564.

Reviewers' comments:

Reviewer's Responses to Questions

**Comments to the Author**

1. Is the manuscript technically sound, and do the data support the conclusions?

Reviewer #1: Yes

Reviewer #2: Yes

2. Has the statistical analysis been performed appropriately and rigorously? 

Reviewer #1: Yes

Reviewer #2: Yes

3. Have the authors made all data underlying the findings in their manuscript fully available?

Reviewer #1: Yes

Reviewer #2: Yes

4. Is the manuscript presented in an intelligible fashion and written in standard English?

Reviewer #1: Yes

Reviewer #2: Yes

5. Review Comments to the Author

Reviewer #1: This paper presents a thoroughly work of manual annotations on a multi-layered dataset of open peer review texts at the sentence level. The authors’ dataset has the potential to pose as a benchmark experimental testbed for automated systems to leverage on current NLP state-of-the-art techniques to address different issues with peer review quality, leading to more transparency and trust on of scientific research validation.

It was interesting to read the paper, I only have minor comments.

Minor comments.

Line 17: please correct “aggravate the problem” to “aggravates the problem”. Please re-read the manuscript to correct some of these typos. Microsoft Word might suggest corrections, that LateX for example might miss. I suggest a copy paste of the text generated without line numbers from pdf to word.

Line 119: in this section the authors keep repeating the term “Authors”. In order not to sound repetitive, the authors might restructure some of the sentences, where possible or adequate, such as, eg replacing “Authors in this paper [45] explored the dubious connection between peer review and quality” by “The dubious connection between peer review and quality was explored in [45]”.

Line 184-186: the abbreviation “ML” was not defined anywhere in the paper. Before using an abbreviation, please define it the first time it is mentioned: machine learning (ML). The same applied to “ABS”, “MET” etc. Although the abbreviations are defined in the tables, if they are mentioned in the text, they should be defined since the reader may not be acquainted with those.

Line 490: What is the criterion for reporting F1 and not recall or both? If F1 measure is the most suited, this should be explained. For example, it is suited for imbalanced datasets.

Line 548: please correct “are not available” to “is not available” (similar to comment for Line 17).

Correct “positive” in the legend of figures in Appendix.

Reviewer #2: The paper addresses the scholarly peer review process and its shortcomings, by devising a gold standard annotated dataset that can help editors assess the quality of a given review based on a series of predefined criteria, on four different levels of analysis. The dataset of c. 17,000 sentences is derived from the 2018 peer review texts produced from one of the main conferences in deep learning (ICLR). Texts were cleaned and tagged for the purposes of using Natural Language Processing (NLP) to classify reviewer statements in four categories: (1) Referred article section; (2) Referred aspect in article; (3) Role in peer review (e.g. critique, suggestion, appreciation); (4) Importance in review text. Very interesting in this regard is the authors analysis of sentiment distribution across different labels.

In the second part of the paper (sections 3 and 4) the authors designed five NLP tasks based on their respective classifications to judge how exhaustive was a given peer-review text. Method used is based on standard machine learning (ML) models, such as BERT. The results for the first four tasks with pre-trained scientific English BERT are above average: between 58% and 65%. Task 5 is the most significant for this judgement overall, as it aims to suggest crucial aspects in a given review or even create an automatic assessment of the text. Its results were poor given the complexity of meta-review preparation.

The paper is generally clearly written. However, language and style could be improved (see my remarks below). Though the results are not cutting edge, the design of the experiment and its implementation using state-of-the-art ML models would lead to better results in the long run. Given necessary fine tuning suggested herein, the article should be accepted for publication.

# introduction

The authors begin from a clear example of their data using three peer review texts of a fictional paper, that was used in a conference tutorial. Given that the authors have assembled a large enough body of data, they should use an example from a solid piece that underwent a rigorous review process as the first example in their study. Otherwise, the whole exercise in judging a good peer review seems rather senseless to the general reader.

# 1 Related Work

This section is adequately comprehensive, but requires some refinement in wording. Especially in the mentioning of actual author last names instead of just the generic "Authors". In the case of research groups one can use first author last name accompanies by et al.

The authors usage of NLP with ML models to study peer review texts indeed seems to be a novelty in the genre of exact science articles. They have written two recent papers on their methods and results. However, analysis with NLP was applied to reviews on student papers at least twice before and should be mentioned in the related work section:

* Xiong, W and Litmaan, D and Schunn, C (2012) Natural language processing techniques for researching and improving peer feedback. Journal of Writing Research 4 (2): 155-176.

* Ramachandran, L., Gehringer, E.F. & Yadav, R.K. (2017) Automated Assessment of the Quality of Peer Reviews using Natural Language Processing Techniques. Int J Artif Intell Educ 27: 534-581. https://doi.org/10.1007/s40593-016-0132-x

# 2.2 Annotation Layers, Schema, and Guidelines

As the authors themselves note, labels used for classifying reviewer statements can sometimes overlap (DFT and CRT; INT and OAL). In other times labels follow fuzzy logic, like differentiating ABS from INT, as well as the use of the DIS and SUG labels. Overall, chosen labels seem to capture the underlying layer of reviewer intent to a good extant. For future iteration of this research I advise that the ontology be exponentially improved through surveys in the scientific community, as well as targeted experimentation on different groups of scholars in a variety of STEM and non-STEM fields.

# 2.6 Sentiment Distribution

Can you explain more about the possible reasons why certain types of labels are positive or negative? this is indeed a very interesting outcome.

# 2.7 Data Quality

The data annotation pipeline is impressive and effective. Especially with regard to minimising errors in the labelling process.

# style, wording and specific issues

As a matter of style its nice that certain important elements were highlighted here and there in the text using italics. But the usage of the italics seem rather excessive overall (e.g. ll. 28-29, 37-38, 64-66, 161-64), and should be focused for necessary emphasis only.

ll. 32-34: it should be made clearer that this statement is subjective and very much depends on the field of study. Reviews in the humanities and social sciences sometimes touch in depth on very few points only.

l. 56: 'peer review statements' should be well defined either here or somewhere else

l. 58: no need for the '(usually begins with)'.

l. 63: 'roles' in the beginning of the line needs a bit more clarification in this context.

ll. 81-82: this statement is not necessarily accurate. The paper's novelty lies more in the way the data was prepared for specific NLP tasks designed for the peer-review process. In other words the interface between the research questions and the underlying method.

l. 118: should be rephrased, 'brief' cannot be used as a verb.

l. 229: is there a difference between 'phrase-level' annotations and 'sentence-level' annotations? Unclear.

6. PLOS authors have the option to publish the peer review history of their article (what does this mean?). If published, this will include your full peer review and any attached files.

Reviewer #1: **Yes: **Marta Fernandes

Reviewer #2: No

---

## [Author Response · Author response to Decision Letter 0]

20 Aug 2021

To,

The Editor, Reviewers,

PLOS ONE

Date: July 6, 2021

Subject: Addressing Editor/Reviewer Comments for PONE-D-20-36706

Dear Editor, Reviewers,

Many thanks to you for your helpful feedback. We are sorry that it took so long for us to revise. Majority of our authors and their families were suffering from Covid-related impact in India. We have now tried to address all your concerns to the best of our abilities. Kindly help us with your further concerns (if any) to improvise our paper. Below we are providing our responses to the comments and actions that we took to address them. The responses are highlighted in blue.

Thank you. Kind regards,

Authors of PONE-D-20-36706

Peer Review Analyze: A Novel Benchmark Resource for Computational Analysis of Peer Reviews

Tirthankar Ghosal, Sandeep Kumar, Prabhat Bharti, Asif Ekbal

PONE-D-20-36706

Peer Review Analyze: A Novel Benchmark Resource for Computational Analysis of Peer Reviews

PLOS ONE

Dear Dr. Ghosal,

Thank you for submitting your manuscript to PLOS ONE. After careful consideration, we feel that it has merit but does not fully meet PLOS ONE’s publication criteria as it currently stands. Therefore, we invite you to submit a revised version of the manuscript that addresses the points raised during the review process.

Included

Included

Included

We look forward to receiving your revised manuscript.

Kind regards,

Lianmeng Jiao

Academic Editor

PLOS ONE

Journal Requirements:

We have now followed the Plos One template available at https://journals.plos.org/plosone/s/latex

2.Thank you for stating the following in the Acknowledgments Section of your manuscript:

"The first author thanks Visvesvaraya Ph.D. Fellowship Award (# VISPHD-MEITY-2518) from the

Ministry of Electronics and Information Technology (MEITY), Government of India. The fourth

author is also a recipient of the Visvesvaraya Young Faculty Award and acknowledges Digital India

Corporation, MEITY for supporting this research."

We have removed the acknowledgment section from the manuscript. Kindly include the following text on the online submission form:

"The first author thanks Visvesvaraya Ph.D. Fellowship Award (# VISPHD-MEITY-2518) from the Ministry of Electronics and Information Technology (MEITY), Government of India. The fourth author is also a recipient of the Visvesvaraya Young Faculty Award and acknowledges Digital India Corporation, MEITY for supporting this research."

3. Please upload a copy of Supporting Information S1: Dataset and Codes which you refer to in your text in line 564.

All the codes and data are available at https://figshare.com/s/203d1c647e69479ac694 Hence we include the above web link for Supporting Information S1

Reviewers' comments:

Reviewer's Responses to Questions

Comments to the Author

1. Is the manuscript technically sound, and do the data support the conclusions?

Reviewer #1: Yes

Reviewer #2: Yes

2. Has the statistical analysis been performed appropriately and rigorously? 

Reviewer #1: Yes

Reviewer #2: Yes

3. Have the authors made all data underlying the findings in their manuscript fully available?

Reviewer #1: Yes

Reviewer #2: Yes

4. Is the manuscript presented in an intelligible fashion and written in standard English?

Reviewer #1: Yes

Reviewer #2: Yes

5. Review Comments to the Author

Reviewer #1: This paper presents a thoroughly work of manual annotations on a multi-layered dataset of open peer review texts at the sentence level. The authors’ dataset has the potential to pose as a benchmark experimental testbed for automated systems to leverage on current NLP state-of-the-art techniques to address different issues with peer review quality, leading to more transparency and trust on of scientific research validation.

It was interesting to read the paper, I only have minor comments.

Thank you dear reviewer, for your encouraging comments.

Minor comments.

Line 17: please correct “aggravate the problem” to “aggravates the problem”. Please re-read the manuscript to correct some of these typos. Microsoft Word might suggest corrections, that LateX for example might miss. I suggest a copy paste of the text generated without line numbers from pdf to word.

We have fixed the problem and thoroughly checked our text for grammatical inconsistencies with Grammarly.

Line 119: in this section the authors keep repeating the term “Authors”. In order not to sound repetitive, the authors might restructure some of the sentences, where possible or adequate, such as, eg replacing “Authors in this paper [45] explored the dubious connection between peer review and quality” by “The dubious connection between peer review and quality was explored in [45]”.

We have now revised the content as per suggestion.

Line 184-186: the abbreviation “ML” was not defined anywhere in the paper. Before using an abbreviation, please define it the first time it is mentioned: machine learning (ML). The same applied to “ABS”, “MET” etc. Although the abbreviations are defined in the tables, if they are mentioned in the text, they should be defined since the reader may not be acquainted with those.

We have now revised the content as per suggestion.

Line 490: What is the criterion for reporting F1 and not recall or both? If F1 measure is the most suited, this should be explained. For example, it is suited for imbalanced datasets.

Thank you, we have now included the reason for choosing F1 measure in section 5.2 Results

Line 548: please correct “are not available” to “is not available” (similar to comment for Line 17).

Corrected.

Correct “positive” in the legend of figures in Appendix.

Corrected.

Reviewer #2: The paper addresses the scholarly peer review process and its shortcomings, by devising a gold standard annotated dataset that can help editors assess the quality of a given review based on a series of predefined criteria, on four different levels of analysis. The dataset of c. 17,000 sentences is derived from the 2018 peer review texts produced from one of the main conferences in deep learning (ICLR). Texts were cleaned and tagged for the purposes of using Natural Language Processing (NLP) to classify reviewer statements in four categories: (1) Referred article section; (2) Referred aspect in article; (3) Role in peer review (e.g. critique, suggestion, appreciation); (4) Importance in review text. Very interesting in this regard is the authors analysis of sentiment distribution across different labels.

In the second part of the paper (sections 3 and 4) the authors designed five NLP tasks based on their respective classifications to judge how exhaustive was a given peer-review text. Method used is based on standard machine learning (ML) models, such as BERT. The results for the first four tasks with pre-trained scientific English BERT are above average: between 58% and 65%. Task 5 is the most significant for this judgement overall, as it aims to suggest crucial aspects in a given review or even create an automatic assessment of the text. Its results were poor given the complexity of meta-review preparation.

The paper is generally clearly written. However, language and style could be improved (see my remarks below). Though the results are not cutting edge, the design of the experiment and its implementation using state-of-the-art ML models would lead to better results in the long run. Given necessary fine tuning suggested herein, the article should be accepted for publication.

Thank you dear reviewer for your helpful comments. We tried to address them as much as we could. We would look forward to your further suggestions to improve our paper.

# introduction

The authors begin from a clear example of their data using three peer review texts of a fictional paper, that was used in a conference tutorial. Given that the authors have assembled a large enough body of data, they should use an example from a solid piece that underwent a rigorous review process as the first example in their study. Otherwise, the whole exercise in judging a good peer review seems rather senseless to the general reader.

Thank you, dear reviewer. We have now included real examples from our dataset.

# 1 Related Work

This section is adequately comprehensive, but requires some refinement in wording. Especially in the mentioning of actual author last names instead of just the generic "Authors". In the case of research groups one can use first author last name accompanies by et al.

We have now revised the related works section as per suggestion.

The authors usage of NLP with ML models to study peer review texts indeed seems to be a novelty in the genre of exact science articles. They have written two recent papers on their methods and results. However, analysis with NLP was applied to reviews on student papers at least twice before and should be mentioned in the related work section:

* Xiong, W and Litmaan, D and Schunn, C (2012) Natural language processing techniques for researching and improving peer feedback. Journal of Writing Research 4 (2): 155-176.

* Ramachandran, L., Gehringer, E.F. & Yadav, R.K. (2017) Automated Assessment of the Quality of Peer Reviews using Natural Language Processing Techniques. Int J Artif Intell Educ 27: 534-581. https://doi.org/10.1007/s40593-016-0132-x

Thank you dear reviewer for these helpful pointers. We have now included these papers in our literature survey section.

# 2.2 Annotation Layers, Schema, and Guidelines

As the authors themselves note, labels used for classifying reviewer statements can sometimes overlap (DFT and CRT; INT and OAL). In other times labels follow fuzzy logic, like differentiating ABS from INT, as well as the use of the DIS and SUG labels. Overall, chosen labels seem to capture the underlying layer of reviewer intent to a good extant. For future iteration of this research I advise that the ontology be exponentially improved through surveys in the scientific community, as well as targeted experimentation on different groups of scholars in a variety of STEM and non-STEM fields.

Thank you, dear reviewer for this suggestion. We have included this proposal as our future scope of research in the conclusion.

# 2.6 Sentiment Distribution

Can you explain more about the possible reasons why certain types of labels are positive or negative? this is indeed a very interesting outcome.

We have mentioned in Section 3.6

# 2.7 Data Quality

The data annotation pipeline is impressive and effective. Especially with regard to minimising errors in the labelling process.

# style, wording and specific issues

As a matter of style its nice that certain important elements were highlighted here and there in the text using italics. But the usage of the italics seem rather excessive overall (e.g. ll. 28-29, 37-38, 64-66, 161-64), and should be focused for necessary emphasis only.

Thank you for pointing this out. We have tried to minimize the emphasized portions.

ll. 32-34: it should be made clearer that this statement is subjective and very much depends on the field of study. Reviews in the humanities and social sciences sometimes touch in depth on very few points only.

We have now added this information with our assertion in Section 1.

l. 56: 'peer review statements' should be well defined either here or somewhere else

We have included the definition in the designated place in the Introduction section.

l. 58: no need for the '(usually begins with)'.

Removed now.

l. 63: 'roles' in the beginning of the line needs a bit more clarification in this context.

The roles are now explicitly mentioned in the corresponding section.

ll. 81-82: this statement is not necessarily accurate. The paper's novelty lies more in the way the data was prepared for specific NLP tasks designed for the peer-review process. In other words the interface between the research questions and the underlying method.

We have rephrased as per suggestion highlighting the exact contribution of this work.

l. 118: should be rephrased, 'brief' cannot be used as a verb.

Corrected in the related works (Section 2).

l. 229: is there a difference between 'phrase-level' annotations and 'sentence-level' annotations? Unclear.

If there was a compound sentence and the reviewer was commenting on two different aspects/sections with different sentiment polarity, we asked our annotators to do a phrase-level annotation instead at the sentence-level to clearly demarcate which portion of the sentence talks about what aspect/section with their associated sentiment. Clarified now in the additional annotation guidelines (section 3.2.5)

6. PLOS authors have the option to publish the peer review history of their article (what does this mean?). If published, this will include your full peer review and any attached files.

Do you want your identity to be public for this peer review? For information about this choice, including consent withdrawal, please see our Privacy Policy.

Reviewer #1: Yes: Marta Fernandes

Reviewer #2: No

---

## [Decision Letter · Decision Letter 1]

18 Oct 2021

Peer Review Analyze: A Novel Benchmark Resource for Computational Analysis of Peer Reviews

PONE-D-20-36706R1

Dear Dr. Ghosal,

We’re pleased to inform you that your manuscript has been judged scientifically suitable for publication and will be formally accepted for publication once it meets all outstanding technical requirements.

Kind regards,

Lianmeng Jiao

Academic Editor

PLOS ONE

Reviewers' comments:

Reviewer's Responses to Questions

**Comments to the Author**

1. If the authors have adequately addressed your comments raised in a previous round of review and you feel that this manuscript is now acceptable for publication, you may indicate that here to bypass the “Comments to the Author” section, enter your conflict of interest statement in the “Confidential to Editor” section, and submit your "Accept" recommendation.

Reviewer #2: All comments have been addressed

2. Is the manuscript technically sound, and do the data support the conclusions?

Reviewer #2: Yes

3. Has the statistical analysis been performed appropriately and rigorously? 

Reviewer #2: Yes

4. Have the authors made all data underlying the findings in their manuscript fully available?

Reviewer #2: Yes

5. Is the manuscript presented in an intelligible fashion and written in standard English?

Reviewer #2: Yes

6. Review Comments to the Author

Reviewer #2: The authors have properly responded to the requested revision and have improved their presentation. They have adequately referred to previous studies that were not mentioned before. Annotation layers implemented the suggested clarifications, and style and language have been improved. I recommend to publish the article.

7. PLOS authors have the option to publish the peer review history of their article (what does this mean?). If published, this will include your full peer review and any attached files.

Reviewer #2: **Yes: **Shai Gordin

---

## [Editor Report · Acceptance letter]

3 Jan 2022

PONE-D-20-36706R1 

Peer Review Analyze: A Novel Benchmark Resource for Computational Analysis of Peer Reviews 

Dear Dr. Ghosal:

I'm pleased to inform you that your manuscript has been deemed suitable for publication in PLOS ONE. Congratulations! Your manuscript is now with our production department. 

Kind regards, 

on behalf of

Dr. Lianmeng Jiao 

Academic Editor

PLOS ONE